# Synaptotagmin 1 and Synaptotagmin 7 promote MR1-mediated presentation of *Mycobacterium tuberculosis* antigens

**Se-Jin Kim[1,2], Jessie C Peterson[3], Andrew J Olive[4], Fikadu G Tafesse[2], Corinna A Kulicke[1], Elham Karamooz[1,3]\*, David Lewinsohn[1,3]\***

[1]Division of Pulmonary, Allergy, and Critical Care Medicine, Oregon Health and Science University, Portland, United States; [2]Department of Molecular Microbiology and Immunology, Oregon Health and Science University, Portland, United States; [3]VA Portland Health Care System, Portland, United States; [4]Department of Microbiology and Molecular Genetics, Michigan State University, East Lansing, United States

## eLife Assessment

This **useful** study examines the contribution of synaptotagmin 1 and synaptotagmin 7 to metabolite antigen presentation to mucosal-associated invariant T (MAIT) cells; it begins to address a critical gap in our understanding of the antigen presentation mechanisms of these cells. Strengths of the study include the use of Mtb to study the dynamics of antigen presentation to MAIT cells instead of a synthetic antigen. The strength of the evidence to support the conclusion is **solid**.

**\*For correspondence:**
karamooz@ohsu.edu (EK);
lewinsod@ohsu.edu (DL)

**Competing interest:** The authors declare that no competing interests exist.

**Abstract** *Mycobacterium tuberculosis* (Mtb) is an intracellular pathogen that can be sensed by T cells, which are essential for the control of infection. In comparison to viral infections, Mtb antigens are relatively limited and hence, challenging to sample. Specialized antigen presentation pathways enable the presentation of such scarce antigens to CD8$^+$ T cells, which are, thus, uniquely poised to survey intracellular environments. A subset of CD8$^+$ T cells prevalent in the airways, known as mucosal-associated invariant T (MAIT) cells, can be activated through the presentation of Mtb antigens via the major histocompatibility complex class I-related protein 1 (MR1) molecule. Prior work demonstrates that endosomal calcium signaling is critical for MR1-mediated presentation of Mtb-derived antigens. Here, we show that the calcium-sensing trafficking proteins Synaptotagmin (Syt) 1 and Syt7 specifically promote MAIT cell activation in response to Mtb-infected cells. In bronchial epithelial cells, Syt1 and Syt7 localize to late endo-lysosomes and MR1 vesicles. Loss of Syt1 and Syt7 results in enlarged MR1 vesicles and an increased number of MR1 vesicles in close proximity to Mtb-containing vacuoles during infection. This study identifies a specialized pathway in which Syt1 and Syt7 facilitate the translocation of MR1 from Mtb-containing vacuoles, potentially to the cell surface for antigen presentation.

## Introduction

Tuberculosis (TB) is the leading cause of infectious disease mortality worldwide with an estimated 10.8 million infections and 1.25 million deaths in 2023 (*World Health Organization, 2024*). TB is caused by the bacillus *Mycobacterium tuberculosis* (Mtb), an intracellular pathogen that invades and replicates within host cells (*Bermudez and Goodman, 1996*; *Pai et al., 2016*; *Cohen et al., 2018*). Unlike viruses, which hijack host machinery to synthesize viral proteins in the cytosol, Mtb antigens are inherently limited. To overcome this limitation, and because Mtb primarily resides within

membrane-bound phagosomes, the immune system utilizes multiple pathways to sample distinct subcellular compartments (*Armstrong and Hart, 1971*; *Jordao et al., 2008*; *Moreira et al., 1997*). CD8⁺ T cells, for instance, can recognize Mtb-derived antigens and eliminate infected cells through specialized antigen processing and presentation mechanisms in myeloid cells as well as the airway epithelium (*Bermudez and Goodman, 1996*; *Cohen et al., 2018*; *Harriff et al., 2014*; *Rozot et al., 2013*). Specifically, a subset of CD8⁺ T cells prevalent in the airways, known as mucosal-associated invariant T (MAIT) cells, detects intracellular infections via the major histocompatibility complex (MHC) class I-related protein 1 (MR1) molecule (*Kjer-Nielsen et al., 2012*; *Corbett et al., 2014*; *Gold et al., 2010*; *Le Bourhis et al., 2010*). MAIT cells rapidly respond to infections, produce proinflammatory cytokines upon activation, and promote antimicrobial activity to control infection (*Gold et al., 2010*; *Le Bourhis et al., 2010*; *Meermeier et al., 2022*). Therefore, understanding the specialized processing of Mtb antigens and their presentation by the MR1 molecule could contribute toward the development of effective vaccines and targeted therapies.

CD8⁺ T cells recognize intracellular antigens presented by HLA-Ia and HLA-Ib. While HLA-Ia activates peptide-specific T cells restricted by a specific allele, HLA-Ib presents non-peptide antigens and interacts with donor-unrestricted T cells that recognize a broad spectrum of antigens (*Godfrey et al., 2015*). MR1 is an evolutionarily conserved HLA-Ib molecule widely expressed in nucleated cells (*Tsukamoto et al., 2013*; *Riegert et al., 1998*). Unlike other HLA-Ib molecules, MR1 presents small molecule metabolites derived from the microbial riboflavin biosynthesis pathway, including those from pathogens such as Mtb, *Salmonella enterica* serovar Typhimurium, and *Escherichia coli* (*Kjer-Nielsen et al., 2012*; *Corbett et al., 2014*; *Chengalroyen, 2024*). Although 5-(2-oxopropylideneamino)-6-d-ribitylaminouracil (5-OP-RU) is the most potent MR1 ligand identified, recent studies show that MR1 presents diverse ligands including those beyond the riboflavin pathway (*Harriff et al., 2018*; *Krawic et al., 2024*; *Meermeier et al., 2016*; *Keller et al., 2017*; *Ito et al., 2024*; *Lepore et al., 2017*; *Salio et al., 2020*; *McInerney et al., 2024*; *Vacchini et al., 2024*; *Chancellor et al., 2025*; *Awad et al., 2025*; *Matsuoka et al., 2023*). For example, MAIT cells recognize infection with *Streptococcus pyogenes*, a microbe that cannot produce riboflavin (*Meermeier et al., 2016*). Additionally, MR1 can present host-derived bile acid metabolites, synthetic ligands, vitamin B6-derived compounds, nucleobase adducts, cigarette smoke components, derivatives of phenylpropanoids, and drug metabolites, leading to MR1-restricted T cell activation (*Keller et al., 2017*; *Ito et al., 2024*; *Salio et al., 2020*; *McInerney et al., 2024*; *Vacchini et al., 2024*; *Chancellor et al., 2025*; *Awad et al., 2025*; *Matsuoka et al., 2023*). These findings highlight the diversity and abundance of MR1 ligands beyond what was previously understood.

The processing and presentation of MR1 ligands depend on the intracellular source of antigens. At steady state, MR1 is minimally expressed on the cell surface and predominantly localizes to the endoplasmic reticulum (ER) and late endosomes (*Harriff et al., 2016*; *McWilliam et al., 2016*; *Huang et al., 2008*). Upon binding to small molecule ligands such as 5-OP-RU or acetyl-6-formylpterin (Ac-6-FP), which form a covalent bond with MR1, the ligand-loaded MR1 complex exits the ER and traffics to the cell surface (*McWilliam et al., 2016*; *McWilliam et al., 2020*). However, it remains unclear whether this pathway accounts for the processing and presentation of antigens derived from intracellular microbes such as Mtb. Previous work from our group demonstrates the role of endosomal proteins in MR1 trafficking and differences in the presentation of intracellular Mtb antigens compared to exogenous antigens (*Harriff et al., 2016*; *Huber et al., 2020*; *Karamooz et al., 2019*). Moreover, recent findings show that inhibition of endosomal calcium release via two-pore channels specifically reduces MR1 presentation of Mtb antigens, but not exogenous antigens (*Karamooz et al., 2025*). Thus, endosomal calcium signaling may serve as a key molecular signal in MR1 antigen processing and presentation of intracellular pathogens.

Multiple studies show that local calcium signaling influences vesicular trafficking, including the secretion of lysosome-related organelles and retrograde transport between cellular compartments (*Patel, 2015*; *Ruas et al., 2015*; *Davis et al., 2012*). As calcium-sensing endosomal trafficking proteins, Synaptotagmins (Syts) are potential downstream effectors of endosomal calcium release that localize to vesicles and mediate membrane fusion events (*Chapman, 2002*; *Südhof, 2013*). Syt1, which is highly enriched in neurons, facilitates vesicle exocytosis for neurotransmitter release (*Brose et al., 1992*; *Geppert et al., 1994*). In contrast, Syt7 is broadly expressed and localizes to late endosomes and lysosomes in macrophages and dendritic cells, where it mediates the translocation

of MHC class II molecules from late endosomes to the plasma membrane (*MacDougall et al., 2018*; *Czibener et al., 2006*; *Becker et al., 2009*). Syts act as part of the soluble *N*-ethylmaleimide sensitive factor attachment protein receptor (SNARE) complex, which includes vesicle-associated membrane protein (VAMP), Syntaxin, and synaptosome-associated protein. The interaction among these SNARE proteins is essential for membrane fusion and intracellular membrane trafficking (*Südhof, 2013*; *Scheller, 2013*; *Bai et al., 2004*). These studies suggest that Syts may act as downstream effectors of localized calcium release. The formation of SNARE complexes facilitates vesicle fusion within endosomal compartments or with the plasma membrane, highlighting their potential role in MR1 antigen presentation.

Here, we demonstrate that Syt1 and Syt7 specifically mediate MR1 presentation of Mtb-derived antigens. Loss of Syt1 and Syt7 disrupts cellular distribution and trafficking of MR1 during intracellular Mtb infection, hindering the translocation of the MR1 molecule from Mtb-containing vacuoles, potentially, to the cell surface for antigen presentation. This study highlights a novel role of Syt1 and Syt7 in MR1 trafficking for the presentation of Mtb antigens and provides critical insights into how the immune system samples and presents intracellular microbes.

## Results

### Syt1 and Syt7 specifically mediate MR1 presentation of intracellular Mtb in epithelial cells

Based on the known function of Syts in vesicle exocytosis and as calcium sensors (*Chapman, 2002*; *Südhof, 2013*), we hypothesized that Syts play a role in MR1-dependent antigen presentation. Among 17 Syt isoforms, Syt1 and Syt7 are the most widely studied Syts (*MacDougall et al., 2018*). We first defined the expression of Syt1 and Syt7, two calcium-sensing Syts, in human bronchial epithelial BEAS-2B cells as well as differentiated monocytic THP-1 cells. Syt1 and Syt7 transcripts were expressed in both cell types (*Figure 1a*). While we previously found that small-interfering RNA (siRNA) knockdown of Syt7 reduced MR1-mediated presentation of Mtb in BEAS-2B cells (*Karamooz et al., 2025*), we sought to further define the role of these proteins in the presentation of mycobacterial antigens. As a result, we generated Syt1 and Syt7 knockout (KO) BEAS-2B cells using a lentiviral CRISPR/Cas9 system. We confirmed the gene knockout efficiency in clonal cell lines through Sanger sequencing and used the Inference of CRISPR Editing (ICE) tool to calculate editing efficiency and percent indel distribution (*Conant et al., 2022*). The editing efficiency was 97% for Syt1 with –1 and –21 indel deletions. For Syt7, the editing efficiency was 96% with –14 and –15 indel deletions (*Figure 1b*). There was no observed cell toxicity in these clonal cell lines. Then, to determine the functional role of Syt1 and Syt7 in antigen presentation, we measured IFN-γ release from human T cell clones co-cultured with Mtb-infected BEAS-2B cells. We found that both Syt1 and Syt7 KO BEAS-2B cells showed a decrease in MR1 presentation of Mtb (*Figure 1c*). To confirm that the degree of intracellular infection was not impacted by genetic deletion of Syt1 and Syt7, we used an HLA-B45-restricted T cell clone specific for the Mtb peptide CFP10$_{2-9}$. Here, we did not observe any differences in the presentation of Mtb-derived antigen. Also, as controls for exogenously delivered antigens, there were no changes in presentation of *Mycobacterium smegmatis* (Msmeg) supernatant or the CFP10$_{2-9}$ peptide to the MAIT cell or the HLA-B45-restricted T cell clone, respectively (*Figure 1d*). These findings show that genetic deletion of Syt1 and Syt7 specifically inhibited MR1 presentation of intracellular Mtb while leaving HLA-Ia (HLA-B45) presentation and presentation of exogenous antigens intact.

To further investigate if Syt1 and Syt7 contribute to an MR1 endosomal pathway involving recycling endosomes, we used a prodrug of 5-amino-6-D-ribitylaminouracil (5-A-RU) as an antigen (*Lange et al., 2020*). 5-A-RU is a precursor of 5-OP-RU, a highly potent MR1 ligand that is unstable without the formation of an MR1–antigen complex. While 5-A-RU is also unstable, a recently developed 5-A-RU prodrug exhibits greater stability. This prodrug requires enzymatic cleavage in acidic compartments for loading to occur in recycling endosomes (*Lange et al., 2020*). We found that deletion of Syt1 or Syt7 in BEAS-2B cells did not affect 5-A-RU prodrug presentation (*Figure 1e*). Therefore, Syt1 and Syt7 are selectively involved in MR1 presentation of Mtb-derived antigens discrete from other MR1 endosomal antigen presentation pathways.

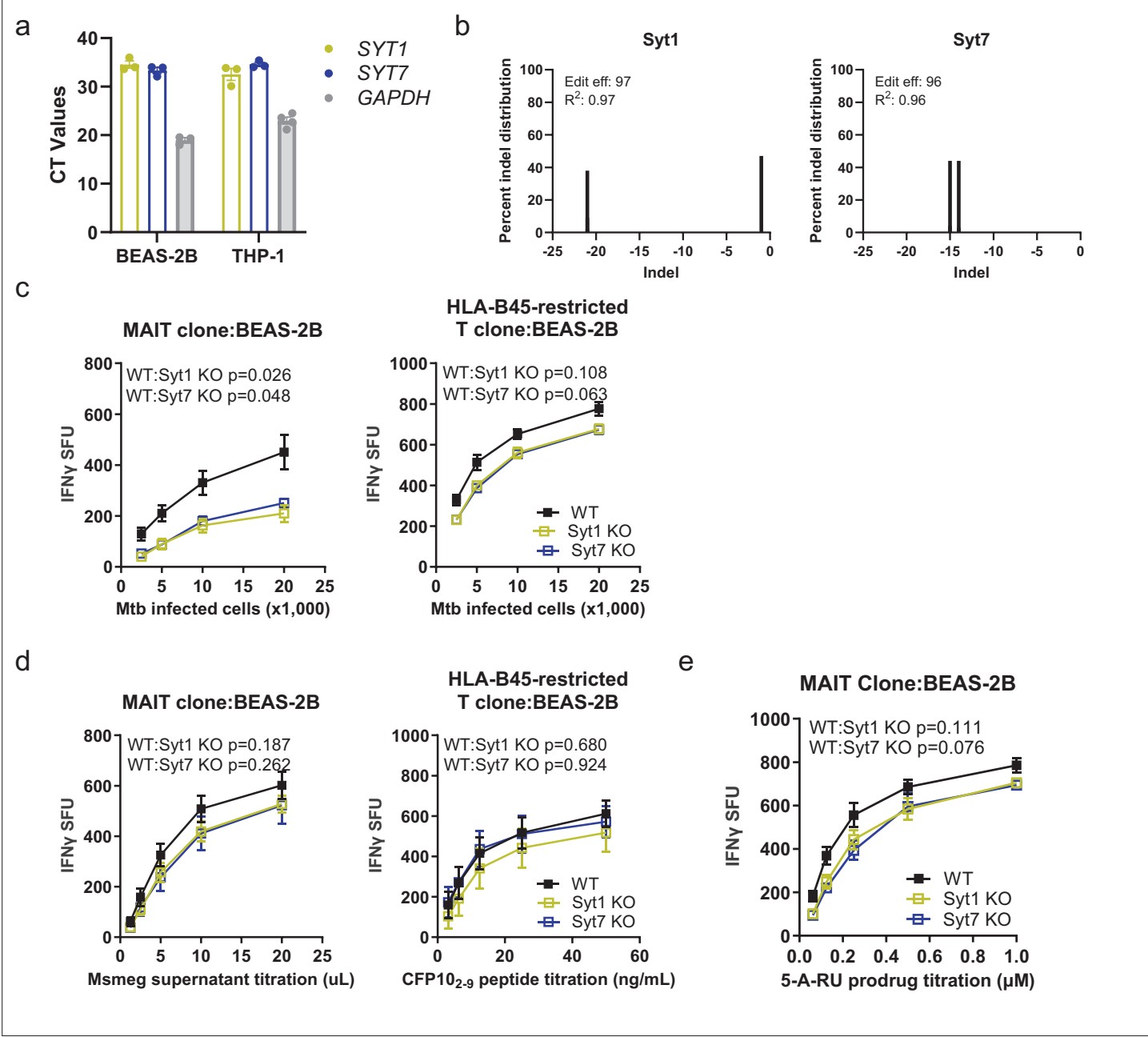

**Figure 1.** Syt1 and Syt7 specifically mediate MR1 presentation of intracellular Mtb. (**a**) Relative gene expression levels of *SYT1*, *SYT7*, and *GAPDH* in BEAS-2B (*n*=3) and PMA-differentiated THP-1 cells (*n*=3-4). (**b**) Genome editing efficiency and percent distribution of individual indels of Syt1 and Syt7 KO BEAS-2B cells as determined by Sanger sequencing and Interference of CRISPR Edits (ICE) (***Conant et al., 2022***) (*n*=1). (**c**) IFN-γ release by T cell clones (MR1- and HLA-B45-restricted) co-cultured with H37Rv Mtb-infected (MOI=8) WT, Syt1 KO, or Syt7 KO BEAS-2B cells, represented as spot forming units (SFU). (**d**) IFN-γ release by T cell clones (MR1- and HLA-B45-restricted) co-cultured with WT, Syt1 KO, or Syt7 KO BEAS-2B cells in the presence of Msmeg supernatant, CFP10$_{2-9}$ peptide, or (**e**) 5-A-RU prodrug, represented as SFU. All data are plotted as mean ± SEM and pooled from *n*=3 independent experiments. For (**c–e**), the means of technical duplicates were pooled, and non-linear regression analysis of pairwise comparison to WT on best-fit values of top and EC$_{50}$ was used to calculate p-values by extra sum-of-squares *F* test. A p value of <0.05 was considered statistically significant.

The online version of this article includes the following source data for figure 1:

**Source data 1.** Source data corresponding to *Figure 1*.

## Syt1 and Syt7 do not affect Mtb uptake and growth

Prior publications have demonstrated that Syt7 can facilitate phagocytosis in macrophages but do not affect phagocytic ability in dendritic cells (*Czibener et al., 2006*; *Becker et al., 2009*). To examine whether Syt1 and Syt7 play a role in the uptake of Mtb in bronchial epithelial cells, we used an auxotrophic strain of Mtb (AuxMtb). AuxMtb is an attenuated strain of the virulent H37Rv Mtb that retains the region of difference 1 (RD1) locus (*Hondalus et al., 2000*; *Jain et al., 2014*). RD1 encodes for ESAT-6 and CFP-10, which create pores and lyse phagosomes as in H37Rv Mtb. We further modified AuxMtb into a live/dead reporter strain, mEmeraldRFP-AuxMtb, which constitutively expresses mEmerald and tetracycline-inducible RFP (*Martin et al., 2012*). To assess uptake, we measured the percentage of live GFP$^+$ cells after overnight infection (*Figure 2a, b*). There were no significant differences in AuxMtb uptake between Syt1 and Syt7 KO and wild-type BEAS-2B cells (*Figure 2c*). To determine the viability of intracellular Mtb, we conducted colony-forming unit (CFU) assays in lysed cells after overnight infection. As shown in *Figure 2d*, we observed no statistical differences in Mtb viability and growth. Lastly, we tested whether there was a change in MR1 expression in Syt1 and Syt7 KO cells, and we found no differences in *MR1* transcripts by RT-qPCR compared to wild-type cells (*Figure 2e*). These findings suggest that Syt1 and Syt7 influence antigen presentation independently of effects on Mtb uptake, Mtb growth, or *MR1* transcript levels.

## Syt1 and Syt7 also mediate MR1 presentation of Mtb in THP-1 cells

We next sought to determine whether the roles of Syt1 and Syt7 are generalizable to professional antigen-presenting cells. As Syt1 and Syt7 are expressed in differentiated monocytic THP-1 cells (*Figure 1a*), we generated Syt1 and Syt7 KO THP-1 cells using the CRISPR/Cas9 system. We confirmed knockout efficiency using Sanger sequencing and the ICE analysis tool. The editing efficiency was 95% for Syt1 with –1 and –20 indel deletions and 100% for Syt7 with –16 indel deletion (*Figure 3a*). To identify the functional role of Syt1 and Syt7 in macrophages for MR1 presentation, we differentiated Syt1 and Syt7 KO THP-1 cells with phorbol 12-myristate 13-acetate (PMA) and measured T cell-dependent IFN-γ release. We found that the genetic deletion of Syt1 and Syt7 in THP-1 cells also resulted in a decrease in MR1 presentation of Mtb, while we observed no changes in the presentation of Msmeg supernatant compared to wild-type cells (*Figure 3b, c*). Due to HLA mismatch, we were not able to include controls for HLA-Ia and CFP10$_{2-9}$ peptide. However, these findings indicate that Syt1 and Syt7 also specifically promote MR1 presentation of Mtb and not exogenous antigen in professional antigen-presenting cells.

## Syt11 and ER-associated Esyt1 and Esyt2 do not selectively affect MR1 presentation of Mtb

To test the hypothesis that Syt1 and Syt7 specifically mediate MR1 presentation of Mtb, we investigated additional calcium-related trafficking proteins (*Wolfes and Dean, 2020*; *Saheki and De Camilli, 2017*). Syt11 is a non-calcium-sensing Syt because it does not contain active calcium binding sites (*Wolfes and Dean, 2020*). It localizes to recycling endosomes and primarily inhibits endocytosis in neurons (*Wang et al., 2016*). Given the extensive study of the ER pathway in MR1 presentation and the role of the ER as the primary calcium storage site (*McWilliam et al., 2016*; *McWilliam et al., 2020*; *Schwarz and Blower, 2016*), we examined the role of ER-associated calcium-sensing proteins Esyt1 and Esyt2, which facilitate ER-plasma membrane tethering (*Giordano et al., 2013*). We found that Syt11, Esyt1, and Esyt2 were highly expressed in BEAS-2B cells (*Figure 4a*). To determine their roles in MR1 presentation, we transfected BEAS-2B cells with siRNAs targeting *SYT11*, *ESYT1*, or *ESYT2* and confirmed over 80% reduction in mRNA transcripts (*Figure 4b*). Co-culture with human T cell clones and measurement of T cell-dependent IFN-γ release revealed three distinct phenotypes. Syt11 knockdown increased MR1 presentation of Msmeg supernatant but had no effect on MR1 presentation of Mtb (*Figure 4c*). In contrast, Esyt2 knockdown reduced MR1 presentation of both Mtb and Msmeg supernatant, while Esyt1 knockdown had no effect (*Figure 4d, e*). To test whether there is a decrease in MR1 surface translocation in the presence of a ligand in Esyt2 knockdown, we used Ac-6-FP, a potent MR1 ligand that induces MR1 translocation from the ER to the cell surface (*Eckle et al., 2014*). Treatment with Ac-6-FP resulted in increased MR1 surface stabilization, but MR1 surface level was significantly lower than those observed in missense control (*Figure 4—figure supplement 1a, b*). Interestingly, the impact of Esyt2 knockdown on MR1 presentation was similar to that of Syntaxin 18

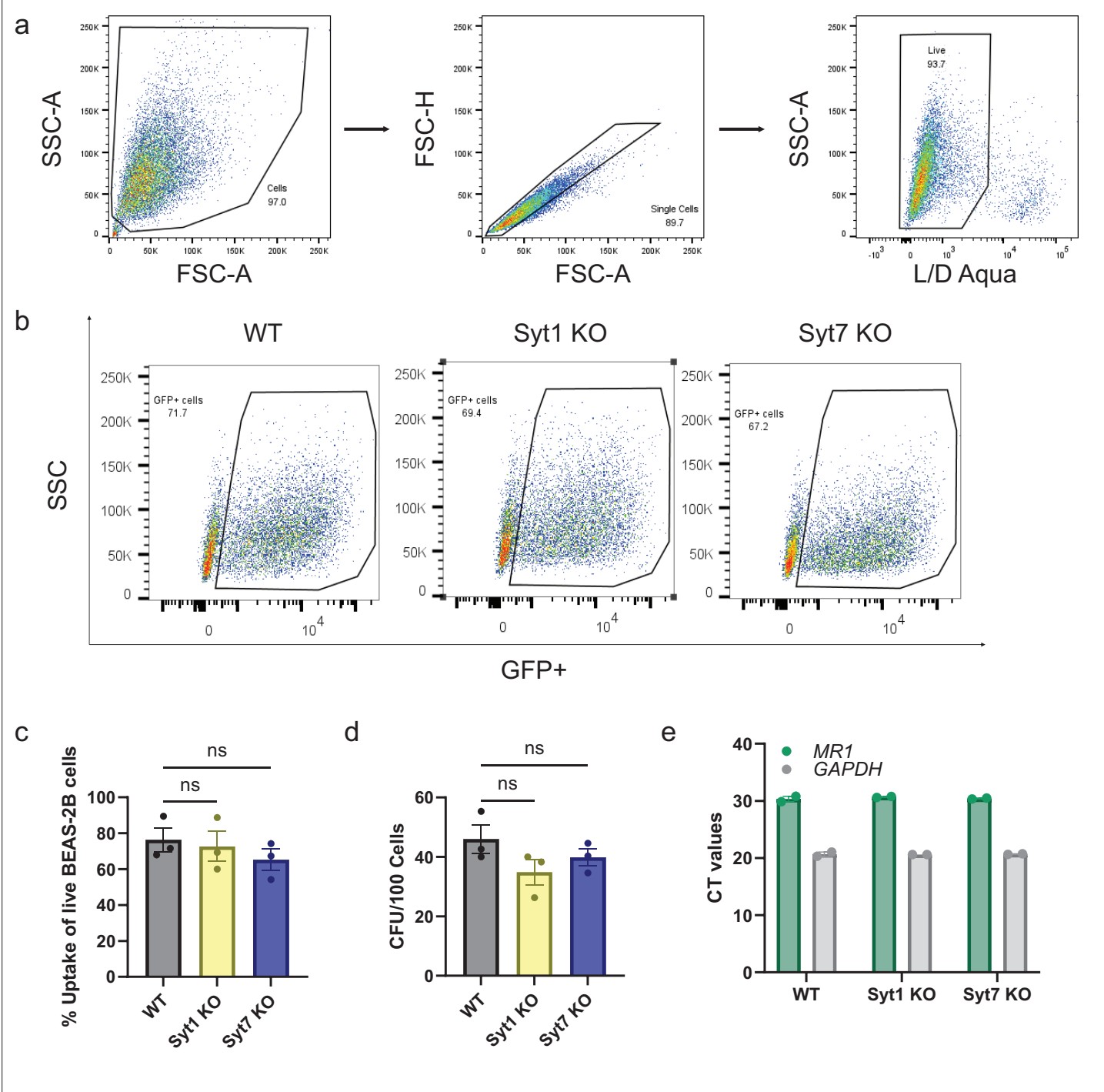

**Figure 2.** Syt1 and Syt7 do not affect Mtb uptake and growth. (**a**) Gating strategy of BEAS-2B cells infected overnight with auxotrophic strain mEmeraldRFP-AuxMtb (MOI=8) by gating on cells, excluding doublets using forward scatter properties, and selecting Live/Dead Near-IR stain negative cells. (**b**) Representative gate on GFP+ population to indicate live BEAS-2B cells infected with mEmeraldRFP-AuxMtb. (**c**) Percent Mtb uptake measured as proportion of live WT, Syt1 KO, or Syt7 KO BEAS-2B cells that are GFP+. (**d**) Colony-forming units (CFU) of H37Rv Mtb in WT, Syt1 KO, or Syt7 KO BEAS-2B cells after overnight infection (MOI=8). (**e**) Relative gene expression levels of *MR1* and *GAPDH* in WT, Syt1 KO, or Syt7 KO BEAS-2B cells. All data are plotted as mean ± SEM. (**c, d**) are pooled from *n*=3 independent experiments and (**e**) is pooled from *n*=2 independent experiments. For (**c, d**), ordinary one-way ANOVA with Dunnett's multiple comparisons test was used to analyze significant differences. ns=not significant (p > 0.05).

The online version of this article includes the following source data for figure 2:

**Source data 1.** Source data corresponding to *Figure 2*.

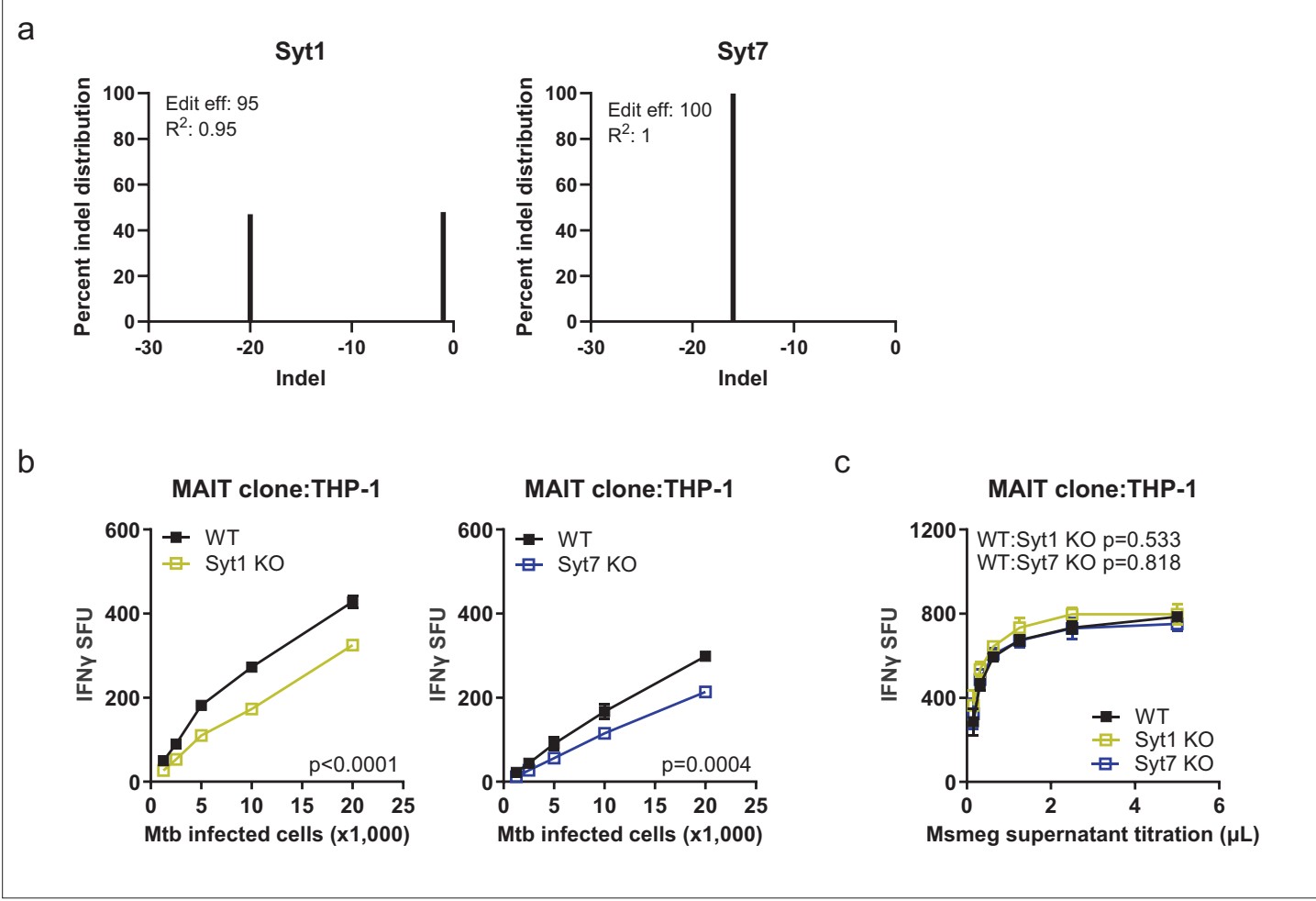

**Figure 3.** Syt1 and Syt7 also mediate MR1 presentation of Mtb in THP-1 cells. (**a**) Genome editing efficiency and percent distribution of individual indels of Syt1 and Syt7 KO THP-1 cells as determined by Sanger sequencing and ICE analysis (*Conant et al., 2022*) (*n*=1). (**b**) IFN-γ release by MAIT cell clones co-cultured with H37Rv Mtb-infected (MOI=1) WT, Syt1 KO, or Syt7 KO THP-1 cells following PMA differentiation, represented as SFU. (**c**) IFN-γ release by MAIT cell clones co-cultured with WT, Syt1 KO, or Syt7 KO THP-1 cells following PMA differentiation in the presence of Msmeg supernatant, represented as SFU. For (**b, c**), the means of technical duplicates were pooled, data were plotted as mean ± SEM and pooled from *n*=3 independent experiments, and non-linear regression analysis of pairwise comparison to WT on best-fit values of top and $EC_{50}$ was used to calculate p-values by extra sum-of-squares *F* test. A p value of <0.05 was considered statistically significant.

The online version of this article includes the following source data for figure 3:

**Source data 1.** Source data corresponding to *Figure 3*.

knockdown (*Harriff et al., 2014*), both of which localize to the ER and influence MR1 presentation of exogenous ligands and intracellular Mtb. Moreover, knockdown of these trafficking proteins did not alter HLA-B45 antigen presentation, indicating specific involvement of Syt11 and Esyt2 in MR1 presentation and distinguishing them from traditional HLA-Ia pathways (*Figure 4c–e*). These findings demonstrate that among the calcium-related trafficking proteins we investigated, Syt1 and Syt7 uniquely contribute to the MR1 presentation of Mtb without affecting exogenous antigen presentation.

## Syt1 and Syt7 localize in late endo-lysosomes and MR1 vesicles

In neurons, Syt1 and Syt7 mainly localize to synaptic vesicles to facilitate membrane fusion (*Chapman, 2002*; *Südhof, 2013*). Other studies show that Syt7 is associated with lysosomes in non-neuronal cells (*Martinez et al., 2000*; *Reddy et al., 2001*). Therefore, we hypothesized that Syt1 and Syt7 will have a similar distribution in lysosomes of bronchial epithelial cells. To define the localization of Syt1 and Syt7, we transfected BEAS-2B cells with RFP-tagged Syt1 and Syt7 and infected them with baculoviruses that deliver constructs to express fluorescent fusion proteins targeted at different subcellular

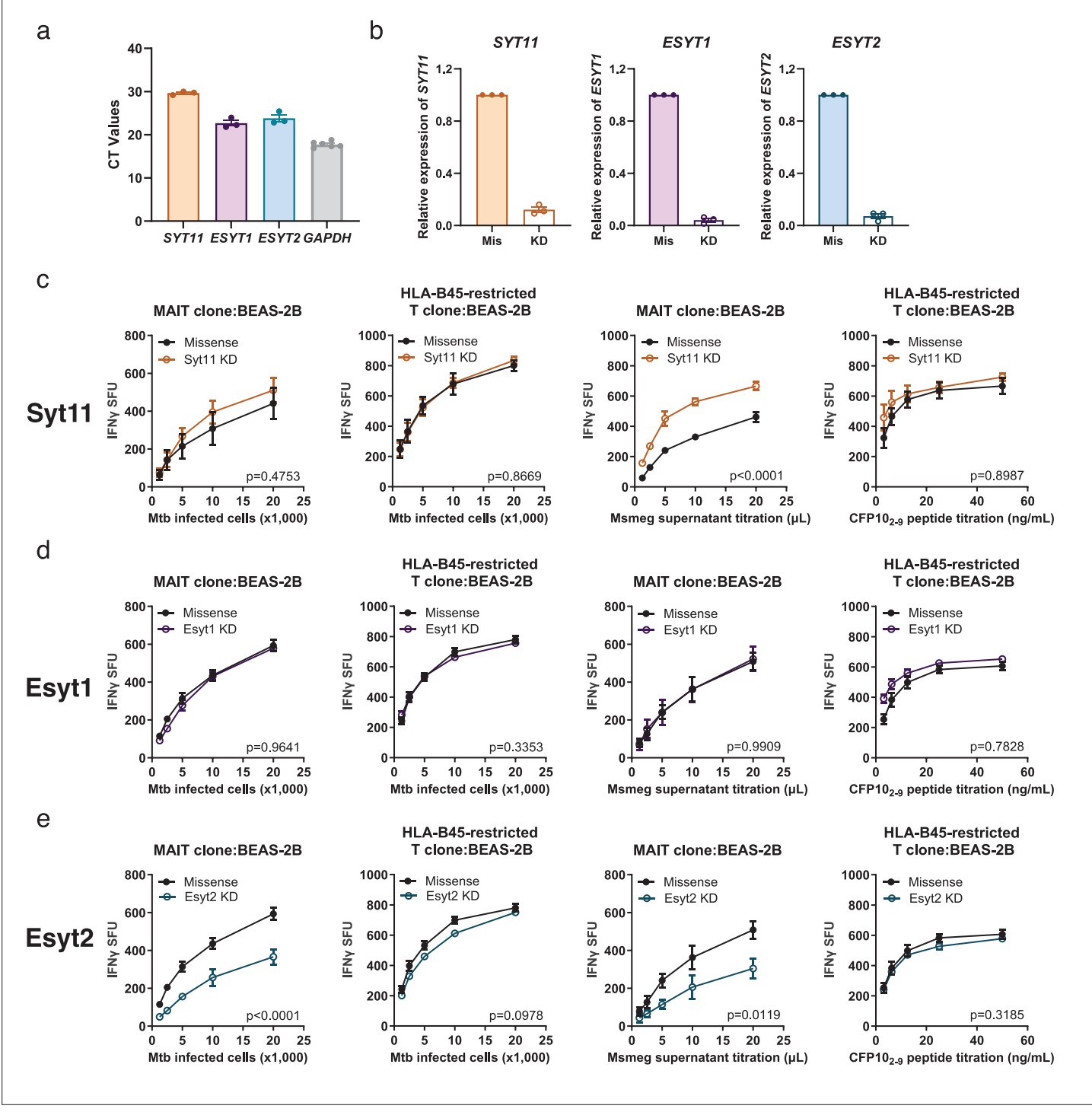

**Figure 4.** Syt11 and ER-associated Esyt1 and Esyt2 do not solely affect MR1 presentation of Mtb. (**a**) Relative gene expression levels of *SYT11* (*n=3*), *ESYT1* (*n=3*), *ESYT2* (*n=3*), and *GAPDH* (*n=6*) in BEAS-2B cells. (**b**) Knockdown efficiency of Syt11, Esyt1, and Esyt2 after 48 hr of knockdown with missense (Mis) or gene-specific (KD) small-interfering RNA (siRNA). IFN-γ release by T cell clones (MR1- and HLA-B45-restricted) co-cultured with BEAS-2B cells following siRNA knockdown of Syt11 (**c**), Esyt1 (**d**), or Esyt2 (**e**). Cells were either infected overnight with H37Rv Mtb (MOI=8) or incubated with exogenously added antigens (Msmeg supernatant and CFP10$_{2-9}$ peptide). IFN-γ release is represented as SFU. All data are plotted as mean ± SEM and pooled from *n=3* independent experiments. Experiments were performed in parallel for (**d, e**). For (**c–e**), the means of technical duplicates were pooled, and non-linear regression analysis comparing best-fit values of top and EC$_{50}$ was used to calculate p-values by extra sum-of-squares *F* test. A p value of <0.05 was considered statistically significant.

*Figure 4 continued on next page*

*Figure 4 continued*

The online version of this article includes the following source data and figure supplement(s) for figure 4:

**Source data 1.** Source data corresponding to *Figure 4*.

**Figure supplement 1.** Esyt2 is important for overall MR1 antigen presentation.

**Figure supplement 1—source data 1.** Source data corresponding to *Figure 4—figure supplement 1*.

compartments (*Kost et al., 2005*). Using fluorescence microscopy, we found that both Syt1 and Syt7 highly co-localized with late endosomes (Rab7a; 60.4% ± 1.7% of vesicles) and lysosomes (LAMP1; 53.0% ± 2.4% of vesicles) compared to early endosomes (Rab5a; 9.3% ± 1.3% of vesicles) (*Figure 5a, b*). Next, we sought to identify the association of Syt1 and Syt7 with MR1 by live-cell imaging. MR1 localizes in the ER and late endo-lysosomes (*Harriff et al., 2016*; *McWilliam et al., 2016*; *Huang et al., 2008*). We used BEAS-2B cells expressing GFP-tagged MR1 to transfect RFP-tagged Syt1 and Syt7. We observed that about 60% of MR1 vesicles co-localized with Syt1 and Syt7 (*Figure 5c, d*), which is

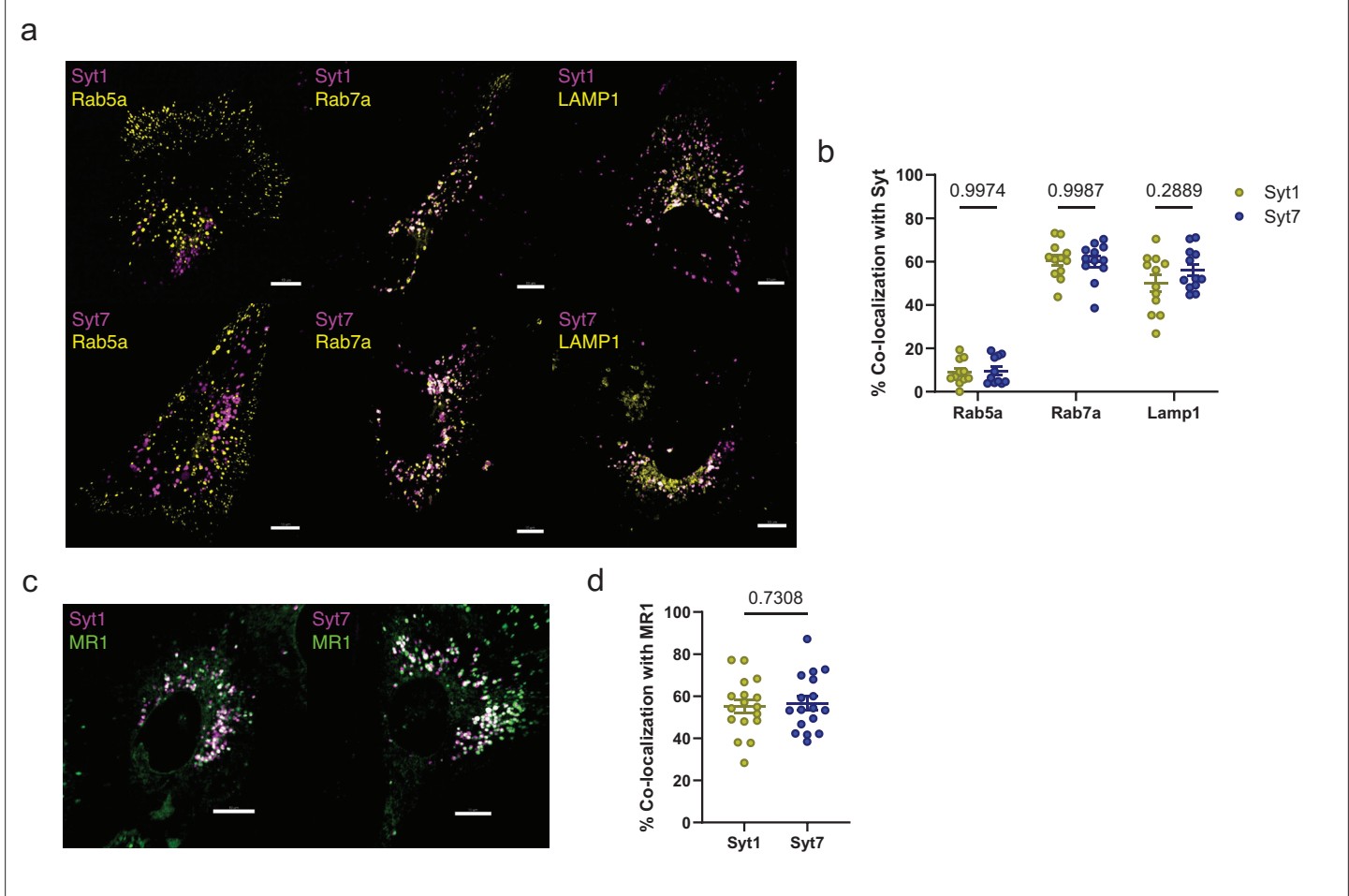

**Figure 5.** Syt1 and Syt7 localize to late endo-lysosomes and MR1 vesicles. (**a**) BEAS-2B cells transfected with Syt1- or Syt7-RFP (magenta) plasmids and incubated with CellLight BacMam 2.0 reagents for Rab5a, Rab7a, and LAMP1 (yellow) overnight. Images are representative of *n*=2 independent experiments (**b**) Percent co-localization of Rab5a (*n*=11), Rab7a (*n*=12), and LAMP1 (*n*=12) with Syt1 or Syt7. Data are pooled from *n*=2 independent experiments and plotted as mean ± SEM. Each dot represents one cell. (**c**) Polyclonal BEAS-2B:TET-MR1GFP (green) cells transfected overnight with Syt1- or Syt7-RFP (magenta) plasmids. Images are representative of *n*=3 independent experiments. (**d**) Percent co-localization of Syt1 or Syt7 with MR1 (*n*=17). Data are pooled from *n*=3 independent experiments and plotted as mean ± SEM. Each dot represents one cell. Two-way ANOVA with Sidak's multiple comparisons test (**b**) and two-tailed unpaired Student's *t*-test (**d**) were used to calculate p-values. A p value of <0.05 was considered statistically significant. All scale bars represent 10 μm.

The online version of this article includes the following source data for figure 5:

**Source data 1.** Source data corresponding to *Figure 5*.

comparable to the extent of co-localization of MR1 with LAMP1 (*Harriff et al., 2014*). These findings suggest a potential role of Syt1 and Syt7 in trafficking MR1 vesicles to or from endo-lysosomes.

## Absence of Syt1 and Syt7 alters MR1 vesicle size in lysosomal compartments

To investigate the role of Syt1 and Syt7 in MR1 vesicular trafficking and cellular distribution, we generated Syt1 and Syt7 KO in BEAS-2B MR1KO cells stably transduced with a doxycycline-inducible MR1-GFP plasmid (BEAS-2B:TET-MR1GFP) (*Kulicke et al., 2025*). We validated knockout efficiency using Sanger sequencing and the ICE tool (*Figure 6a*). Similar to wild-type BEAS-2B cells, Syt1 and Syt7 KO BEAS-2B:TET-MR1GFP cells exhibited a decrease in MR1 presentation of Mtb (*Figure 6b*). Syt7 KO BEAS-2B:TET-MR1GFP cells also demonstrated a small but statistically significant reduction in HLA-B45 presentation of Mtb. However, there was a more pronounced effect in MR1-mediated antigen presentation in Syt7 KO cells compared to WT. We hypothesized that the decreased MR1-mediated presentation of Mtb antigen could be due to decreased MR1 surface translocation in the presence of a ligand. Live-cell imaging showed no differences in MR1 cellular distribution in the presence or absence of Ac-6-FP between WT, Syt1, and Syt7 KO BEAS-2B:TET-MR1GFP cells as MR1 translocated from the ER and vesicles to the cell surface as expected (*Figure 6c*). Similarly, Syt1 and Syt7 KO cells did not alter MR1 surface translocation and HLA-Ia expression in the presence or absence of Ac-6-FP when measured by flow cytometry (*Figure 6d*). These findings suggest that Syt1 and Syt7 are not involved in the ER pathway of MR1 presentation of exogenous ligands and indicate that the reduction in MR1-mediated presentation of Mtb antigen in Syt1 and Syt7 KO cells is independent of ER-mediated MR1 surface translocation.

We further explored MR1 cellular distribution in Syt1 and Syt7 KO BEAS-2B:TET-MR1GFP cells by live-cell imaging. Interestingly, Syt1 and Syt7 KO cells exhibited enlargement of MR1 vesicles and an accumulation of vesicles in close proximity to one other (*Figure 6e*). Quantitative analysis showed a significant increase in the area of larger MR1 vesicles in Syt1 (~60%) and Syt7 (~35%) KO cells compared to WT cells (*Figure 6f*). Given that Syt1, Syt7, and MR1 co-localize with markers associated with late endosomes (Rab7a) and lysosomes (LAMP1) at steady state, we hypothesized that these larger and accumulated MR1 collections localized to lysosomal compartments. To test this, we infected Syt1 and Syt7 KO BEAS-2B:TET-MR1GFP cells with baculoviruses expressing RFP in either early endosomes (Rab5a) or lysosomes (LAMP1) (*Kost et al., 2005*). We found that the larger MR1 vesicles in Syt1 and Syt7 KO cells co-localized more with LAMP1 than Rab5a, similar to WT cells (*Figure 6g, h*). These findings suggest that while Syt1 and Syt7 are not involved in the recruitment of MR1 vesicles to LAMP1+ compartments, they play a critical role in the trafficking of MR1 vesicles from LAMP1+ compartments to other endosomal compartments or to the cell surface for antigen presentation.

## Syt1 and Syt7 mediate trafficking of MR1 vesicles from the Mtb-containing vacuole

To understand whether Syt1 and Syt7 are involved in trafficking MR1 vesicles for antigen presentation during intracellular Mtb infection, we conducted live-cell imaging to visualize Mtb-containing vacuoles. Using Syt1 and Syt7 KO BEAS-2B:TET-MR1GFP cells, we infected the cells with wild-type AuxMtb labeled with Alexa Fluor 555 NHS ester dye. Live-cell imaging showed that MR1 vesicles accumulated near Mtb-containing vacuoles in Syt1 and Syt7 KO cells (*Figure 7a*). Quantitative analysis demonstrated a significant increase in the number of MR1 vesicles within 1 μm of AuxMtb for Syt1 (1.23 ± 0.21) and Syt7 KO (1.28 ± 0.22) cells compared to WT cells (*Figure 7b*). However, there were no significant differences in the total number or average speed of MR1 vesicles between WT, Syt1, and Syt7 KO cells (*Figure 7c*). Furthermore, the surface of MR1 vesicles in Syt1 and Syt7 KO cells showed an approximately fourfold increase in overlap area with Mtb surfaces (*Figure 7d*). To confirm the increased number and surface overlap of MR1 vesicles with Mtb-containing vacuoles of Syt1 and Syt7 KO cells, we quantified MR1 expression within phagolysosomes using flow organellometry. Our lab previously established this method for separating and phenotyping subcellular compartments (*Grotzke et al., 2009*; *Ramachandra et al., 2001*). Flow organellometry enables separation of plasma membrane, ER, lysosomes, and phagosomes in a 27% Percoll gradient. Phagosomes are detected in the last few subcellular fractions (*Grotzke et al., 2009*). We infected WT, Syt1, and Syt7

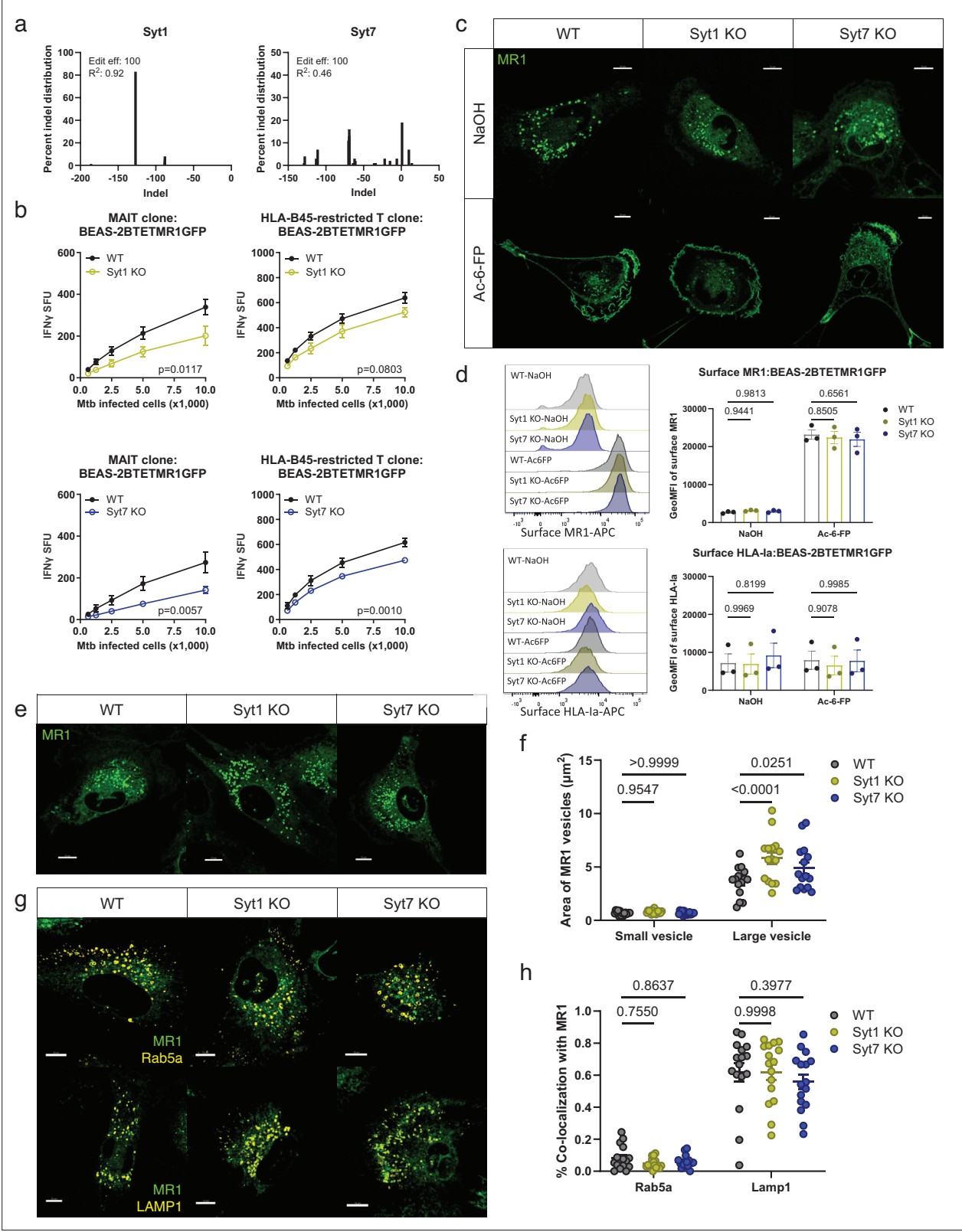

**Figure 6.** Absence of Syt1 and Syt7 alters MR1 vesicle size in lysosomal compartments. Syt1 KO and Syt7 KO were generated in the background of BEAS-2B MR1KO:tetMR1-GFP clone D4 cells. (**a**) Syt1 and Syt7 KO clones were verified by Sanger sequencing and analyzed using the ICE tool (***Conant et al., 2022***) (*n*=1). (**b**) IFN-γ release by T cell clones (MR1- and HLA-B45-restricted) co-cultured with H37Rv Mtb-infected cells (MOI=8) is represented as SFU. The means of technical duplicates were pooled from *n*=3-4 independent experiments, and non-linear regression analysis comparing best-fit values

*Figure 6 continued*

of top and EC$_{50}$ was used to calculate p-values by extra sum-of-squares *F* test. (**c, d**) WT, Syt1 KO, and Syt7 KO BEAS-2B MR1KO:tetMR1-GFP cells were incubated overnight with doxycycline and Ac-6-FP or NaOH (solvent control). Images (**c**) and histograms (d, left) representative of *n*=3 independent experiments with pooled geometric mean fluorescence intensity (GeoMFI) (d, right) of surface MR1 and HLA-Ia expression. (**e**) Representative images of WT, Syt1 KO, and Syt7 KO EAS-2B MR1KO:tetMR1-GFP cells incubated overnight with doxycycline and (**f**) measurement of area of MR1 vesicles, classified into small (1 vesicle) or large (>1 vesicle) vesicles (*n*=15). Each dot represents one cell. Data are pooled from *n*=3 independent experiments. (**g**) Representative images of WT, Syt1 KO, and Syt7 KO BEAS-2B MR1KO:tetMR1-GFP cells incubated overnight with doxycycline and CellLight BacMam 2.0 reagents for Rab5a and LAMP1 (yellow). (**h**) Percent co-localization of Rab5a (*n*=16) and LAMP1 (*n*=16) with MR1 vesicles. Each dot represents one cell. Data are pooled from *n*=4 independent experiments. For (**d, f, h**), p-values were analyzed by two-way ANOVA with Dunnett's multiple comparisons test. All data are plotted as mean ± SEM. A p value of <0.05 was considered statistically significant. All scale bars represent 10 μm.

The online version of this article includes the following source data for figure 6:

**Source data 1.** Source data corresponding to *Figure 6*.

KO BEAS-2B:TET-MR1GFP cells overnight with wild-type AuxMtb labeled with Alexa Fluor 647 NHS ester dye. We performed flow organellometry to separate phagosomes from other subcellular fractions and identified enrichment of Mtb-containing vacuoles in fractions 42–50 (*Figure 7e, f*). When comparing MR1 expression within the fractions containing the highest percentage of Mtb$^+$LAMP1$^+$ vesicles, Syt7 KO cells showed about 25% increase in the geometric mean fluorescence intensity of MR1 compared to WT cells (*Figure 7g*). Although these findings do not determine the exact location of MR1 loading, they support a model whereby Syt1 and Syt7 transport MR1 vesicles, whether loaded or unloaded, from the Mtb-containing vacuoles and to the cell surface for antigen presentation.

## Discussion

CD8$^+$ T cells play a unique role in surveilling intracellular environments. Sampling antigens from intracellular microbes is challenging due to their compartmentalization in membrane-bound organelles, low antigen concentrations, and pathogen immune evasion strategies (*Armstrong and Hart, 1971*; *Jordao et al., 2008*; *Moreira et al., 1997*; *Harriff et al., 2014*). Thus, the immune system utilizes a specialized mechanism to process and present internalized microbes in the context of HLA-Ia in order to elicit strong cytotoxic T cell responses (*Blander, 2018*; *Desjardins, 2019*). Complementing ER-based HLA-Ia-mediated antigen presentation, this highly efficient pathway surveys phagosomes, where foreign antigens are most enriched as they are the first compartment encountered by intracellular microbes. Similarly, HLA-Ib antigen-presenting molecules sense and load unique antigens from different subcellular compartments. For example, HLA-E can process and load antigens in the phagosomes, whereas different CD1 isoforms have distinct antigen sources and loading compartments (*Grotzke et al., 2009*; *Moody and Porcelli, 2003*). Interestingly, the known MR1 ligands are small, hydrophilic, and unstable under physiological conditions, which are unlikely to be easily diffused or transported between subcellular compartments (*Lange et al., 2020*; *Mak et al., 2017*). Accordingly, this study highlights a distinct MR1 endosomal antigen presentation pathway mediated by Syt1 and Syt7 during intracellular Mtb infection and suggests MR1 antigen sampling could occur in Mtb-containing vacuoles.

Although the ER pathway of MR1 presentation is well characterized, this pathway does not fully address processing and presentation of intracellular microbes. McWilliam and colleagues show that MR1 mainly resides in the ER where the loading of exogenous ligand occurs. Small ligands such as 5-OP-RU and Ac-6-FP form a Schiff base bond with lysine 43 of MR1, serving as a molecular signal for MR1 to egress the ER and reach the cell surface (*Corbett et al., 2014*; *McWilliam et al., 2016*). Also, treatment with Brefeldin A results in diminished recognition of *S. enterica* serovar Typhimurium, implying transport of loaded MR1 from the ER to the cell surface or MR1 unable to reach the endosomes for loading. Further study has also identified interactions between MR1 and ER-associated chaperone proteins, such as TAPBPR and tapasin (*McWilliam et al., 2020*). However, these studies primarily use hematopoietic C1R cells in conjunction with small MR1 ligands added to the extracellular culture medium. Therefore, the mechanisms of MR1 processing and presentation in Mtb-infected antigen-presenting cells remain incompletely understood. Further investigation is needed to determine whether this pathway is universal for all intracellular microbes, how it varies across different

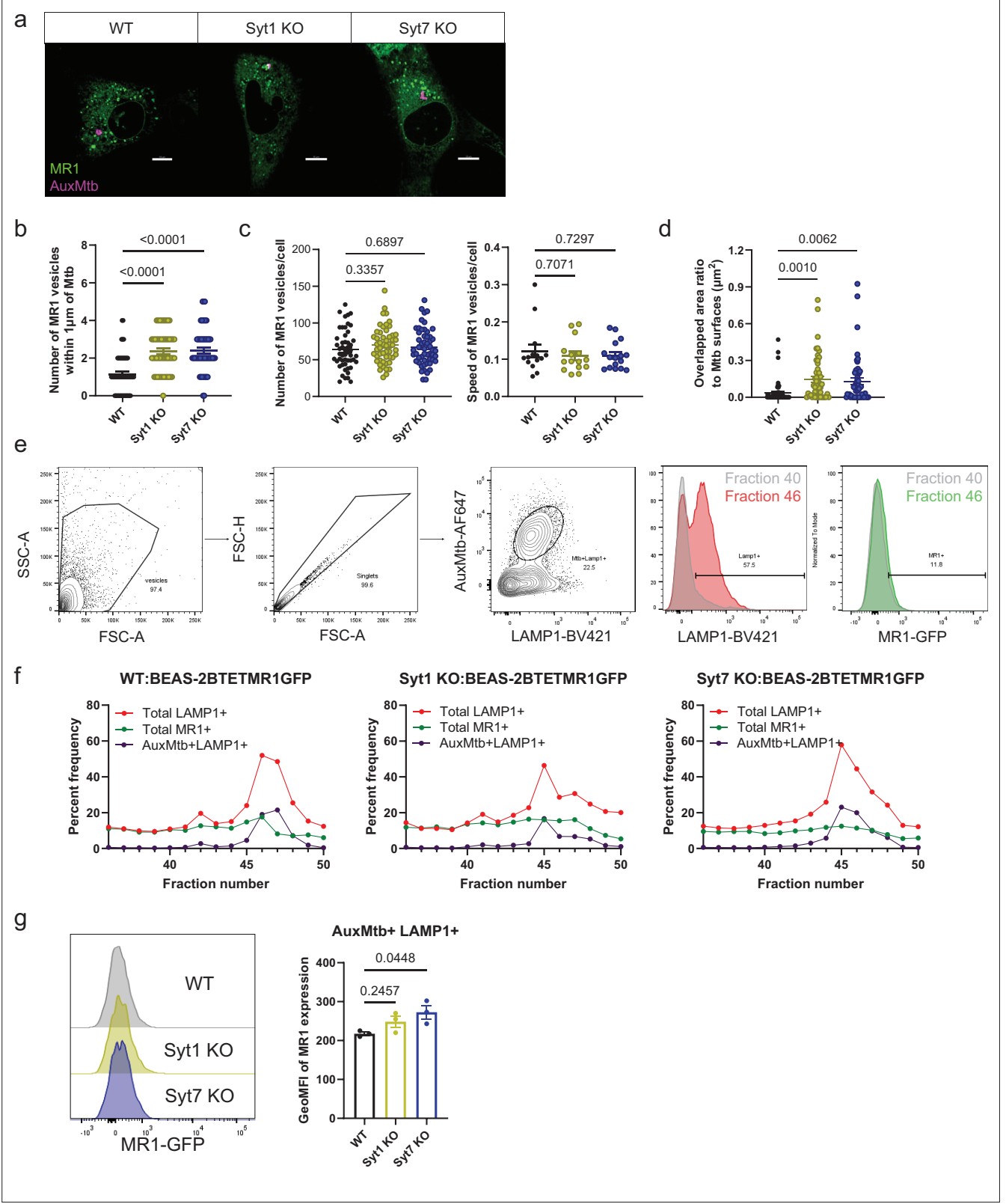

**Figure 7.** Syt1 and Syt7 mediate trafficking of MR1 vesicles from the Mtb-containing vacuole. (**a**) WT, Syt1 KO, and Syt7 KO BEAS-2B MR1KO:tetMR1-GFP (green) cells were infected with AuxMtb (MOI=5) labeled with Alexa Fluor 555 Succinimidyl Ester (AuxMtb-Alexa Fluor555; magenta). Images representative of *n*=3 independent experiments are shown. Scale bars represent 10 µm. (**b**) Number of MR1 vesicles within 1 µm of the center of the AuxMtb surface (*n*=51–53). (**c**) Total number (left, *n*=51–53) and average speed (µm/s) (right, *n*=14–16) of MR1 vesicles. (**d**) Overlapped area ratio of MR1

*Figure 7 continued on next page*

*Figure 7 continued*

to Mtb surfaces (*n*=51–53). For (**b–d**), data are plotted as mean ± SEM and pooled from *n*=8 independent experiments. Each dot represents one cell. p-values were analyzed by a one-way ANOVA with Dunnett's multiple comparisons test. (**e**) WT, Syt1 KO, and Syt7 KO BEAS-2B MR1KO:tetMR1-GFP cells were infected overnight with AuxMtb (MOI=10) labeled with Alexa Fluor 647 Succinimidyl Ester. Mtb-containing vacuoles for each fraction from flow organellometry assay were identified by gating on vesicles, excluding doublets using forward scatter properties, and selecting the AuxMtb⁺LAMP1⁺ population (left). Total Lamp1⁺ and MR1⁺ populations were gated following the same strategy (right). Representative histograms comparing fractions 40 and 46 are shown. (**f**) Graphs representative of *n*=3 independent experiments showing the frequency of total LAMP1⁺, total MR1⁺, and AuxMtb⁺LAMP1⁺ vesicles of all vesicles from fractions 36–50. (**g**) Representative histogram and geometric mean fluorescence intensity (GeoMFI) of MR1 in the subcellular fraction with the highest percentage of AuxMtb⁺LAMP1⁺ vesicles. Data are pooled from *n*=3 independent experiments and plotted as mean ± SEM. p-values were analyzed by a one-way ANOVA with Dunnett's multiple comparisons test. A p value of <0.05 was considered statistically significant.

The online version of this article includes the following source data for figure 7:

**Source data 1.** Source data corresponding to *Figure 7*.

cell types, and whether specific chaperone proteins assist in antigen loading. These discoveries will extend our understanding of diverse pathways involved in MR1 processing and presentation.

In the endosomal pathway of MR1 presentation, endosomal trafficking proteins and calcium signaling within endosomal compartments facilitate MR1 trafficking and presentation (*Harriff et al., 2016*; *Huber et al., 2020*; *Karamooz et al., 2019*; *Karamooz et al., 2025*; *Kulicke et al., 2024*). Studies investigating the role of endosomal proteins and using exogenously derived antigens and intracellular Mtb infection show multiple MR1 antigen presentation pathways, which are influenced by the type and method of antigen delivery. For instance, VAMP4 knockdown disrupts MR1 presentation of Mtb, while Syntaxin 4 only affects presentation of Msmeg supernatant in bronchial epithelial cells (*Harriff et al., 2016*; *Karamooz et al., 2019*). Additionally, ligand exchange of pre-loaded MR1 occurs in post-ER compartments (*Kulicke et al., 2024*). Moreover, recent evidence shows endosomal calcium signaling as a potential mechanism to detect intracellular infection. Inhibition of endosomal calcium release from two-pore channels specifically decreases MR1 presentation of Mtb in human dendritic cells, setting this pathway apart from other pathways of MR1 antigen presentation (*Karamooz et al., 2025*).

As calcium-sensing endosomal trafficking proteins, Syt1 and Syt7 are potential downstream effectors of endosomal calcium release. Syts are membrane-bound proteins in vesicles that mediate docking and fusion to exquisitely regulate exocytosis of synaptic vesicles (*Chapman, 2002*; *Südhof, 2013*; *Scheller, 2013*). While Syts are extensively studied in presynaptic neurotransmitter release, recent evidence highlights broader roles for Syt1 and Syt7 in epithelial and antigen-presenting cells. Syt1 is expressed in large dense core vesicles of adrenal chromaffin cells, where it regulates hormone release into the bloodstream (*Voets et al., 2001*). Loss of Syt1 in chromaffin cells results in delayed and reduced vesicle exocytosis. Syt7, on the other hand, is broadly expressed across human tissues compared to Syt1 (*MacDougall et al., 2018*). Syt7 localizes to dense lysosomes in rat kidney cells, where its main function is to mediate lysosomal exocytosis (*Martinez et al., 2000*). Further studies expand the role of Syt7 to immune cells, such as bone marrow-derived macrophages and dendritic cells. In macrophages, Syt7 is essential for efficient phagocytosis and the delivery of lysosomes to nascent phagosomes (*Czibener et al., 2006*). In dendritic cells, however, Syt7 deletion does not affect phagocytosis, but significantly delays the translocation of MHC class II to the cell surface (*Becker et al., 2009*). Therefore, we hypothesized that this controlled regulation of exocytosis by Syt1 and Syt7 may play a role in this specialized mechanism of surveilling intracellular microbes by MR1.

The present study supports a unique pathway, in which Syt1 and Syt7 specifically mediate MR1 presentation during intracellular Mtb infection. HLA-Ia presentation of intracellular Mtb as well as presentation of exogenously derived antigens remained intact upon genetic deletion of Syt1 and Syt7. To understand the mechanism, we defined the localization of Syt1 and Syt7 in late endosomes and lysosomes of bronchial epithelial cells. Notably, we did not observe significant differences in Mtb uptake in Syt1 and Syt7 KO cells, similar to previous findings showing that loss of Syt7 does not affect phagocytosis in dendritic cells (*Becker et al., 2009*). However, our Mtb uptake assay involved overnight infections, in which other Syt proteins might compensate for the loss of Syt1 and Syt7 in facilitating phagocytosis. Importantly, the absence of Syt1 and Syt7 in bronchial epithelial cells altered MR1 cellular distribution. Larger MR1 vesicles that accumulated in close proximity to one another were LAMP1⁺, suggesting impaired MR1 vesicle trafficking from lysosomal compartments.

This observation aligned with previous studies showing that Syt7 mobilizes lysosomal membrane to other compartments (*Czibener et al., 2006*). Although Czibener and colleagues showed that Syt7 is recruited to nascent phagosomes in macrophages during zymosan particle uptake, their follow-up studies demonstrated a different role in dendritic cells. Specifically, in mature dendritic cells, Syt7 translocated to the plasma membrane, and its absence delayed translocation of MHC class II to the cell surface (*Becker et al., 2009*). Consistent with these findings, we observed an increase in the number of MR1 vesicles near Mtb-containing vacuoles by fluorescent microscopy and enhanced MR1 expression in Mtb⁺LAMP1⁺ compartments via flow organellometry in Syt1 and Syt7 KO cells. Therefore, we postulate the role of Syt1 and Syt7 in delivering MR1–Mtb antigen complexes from Mtb-containing vacuoles to the cell surface for antigen presentation.

The current work extends our understanding of sampling intracellular microbes through a distinct MR1 endosomal pathway. Inhibition of endosomal calcium release in human dendritic cells and deletion of Syt1 and Syt7 in bronchial epithelial and differentiated monocytic cells all specifically reduce MR1 presentation of Mtb. This suggests that Syt1 and Syt7 act as downstream effectors of endosomal calcium signaling, which facilitate MR1 presentation of Mtb-derived antigens to activate MAIT cells. Moreover, these findings continue to support the existence of distinct MR1 antigen presentation pathways that differ by the type and method of antigen delivery. The endosomal pathway of MR1 presentation involving calcium signaling specifically affects MR1 presentation of Mtb without impacting the ER pathway and other endosomal pathways, reinforcing a specialized mechanism in surveying intracellular microbes. Furthermore, this study expands the role of Syts beyond synaptic vesicle exocytosis, demonstrating their function in antigen presentation during intracellular infection. Our data support a model in which Syt1 and Syt7 mediate the final step in a highly efficient pathway that enables the presentation of MR1–Mtb antigen complexes at the cell surface.

While this study highlights the unique roles of Syt1 and Syt7 in a specialized mechanism of sensing intracellular Mtb, several questions remain. In particular, the potential compensatory roles of other Syt family members and the function of calcium binding domains in antigen presentation warrant further investigation. Follow-up studies could identify other SNARE proteins involved in this pathway. Furthermore, our findings suggest that antigen loading may occur in Mtb-containing vacuoles due to their proximity to and the fact that phagosomes are the initial compartments to acquire intracellular microbes. However, additional investigation is needed to track small molecule Mtb-derived antigens in order to determine the exact location of Mtb antigen loading and explore the presence of chaperone proteins in specific subcellular compartments. Overall, this study establishes a foundation for exploring calcium-sensitive endosomal proteins in antigen presentation and its broader implications in sampling intracellular microbes.

## Methods

### Bacterial strains and cell lines

H37Rv *Mycobacterium tuberculosis* (Mtb) strain, obtained from the American Type Culture Collection (ATCC), was grown in Middlebrook 7H9 Broth supplemented with Middlebrook ADC (BD), 0.05% Tween-80 (OmniPur), and 0.5% glycerol (Thermo Fisher Scientific). Mtb was used from frozen glycerol stocks by passaging 20 times through a tuberculin syringe (BD) with a 27-gauge needle before infection. Multiplicity of infection (MOI) of 8 was used for IFN-γ ELISpot assays. ΔleuD ΔpanCD double auxotroph Mtb (AuxMtb), an attenuated strain of H37Rv Mtb, was gifted by Dr. William Jacobs (*Hondalus et al., 2000*; *Jain et al., 2014*). A modification of AuxMtb was generated to constitutively express the green fluorescent protein mEmerald and the tetracycline (tet)-inducible red fluorescent protein (mEmeraldRFP-AuxMtb) (*Martin et al., 2012*). MOIs of 5 (for fluorescence microscopy), 8 (for Mtb uptake assay), and 10 (for flow organellometry assay) were used. *Mycobacterium smegmatis* (Msmeg, mc² 155 strain) supernatant was obtained by culturing Msmeg for 24 hr in 7H9 broth, passing the supernatant through a 0.22-μm filter, and concentrating it using a 10-kDa Amicon filter (Millipore Sigma). Msmeg supernatant was used from frozen aliquots stored at –80°C.

BEAS-2B and THP-1 cells were obtained from ATCC. BEAS-2B cells were cultured in DMEM (Gibco) supplemented with 10% heat-inactivated fetal bovine serum (GeminiBio) and L-glutamine (Gibco). THP-1 cells were cultured in RPMI 1640 (Gibco) supplemented with 10% heat-inactivated fetal bovine serum and L-glutamine. Polyclonal BEAS-2B:tetMR1-GFP cells were developed by transducing

BEAS-2B cells with tet-inducible MR1-GFP as described previously (*Huber et al., 2020*). BEAS-2B MR1KO:tetMR1-GFP clone D4 was generated by reconstituting a clonal BEAS-2B MR1 KO cell line with a lentiviral vector encoding GFP-tagged MR1A under a tet-inducible promoter (*Narayanan et al., 2020*). The polyclonal cell line was sorted based on GFP expression, and individual clonal lines were generated (*Kulicke et al., 2025*). Due to the constitutive expression of Cas9 and the sgRNA targeting MR1, the MR1 sequence at each insertion of the tetMR1-GFP cassette into the genome was edited. CRISPR editing resulted in a V12I substitution for clone D4, which is phenotypically similar to cell lines over-expressing WT MR1A (*Narayanan et al., 2020*). Of note, bovine and porcine MR1 have an I at this position (*Xiao et al., 2019*). Clone D4 was then used to generate Syt1 and Syt7 KO BEAS-2B MR1KO:tetMR1-GFP cells.

All cell lines were passaged up to 15 times before being discarded. Freezebacks are frozen down during the first 2–3 passages. Cell lines were periodically tested for mycoplasma contamination and confirmed negative. Modification to cell lines was validated. Cells transduced with tet-inducible MR1-GFP were verified by flow cytometry and western blot. Knockout cells were verified by Sanger sequencing and analyzed with the ICE (Synthego) tool (*Conant et al., 2022*).

## Lentiviral-mediated gene knockouts

Lentiviral particles for sgRNAs targeting Syt1 (GGTACCATACTCGGAATTGGG) and Syt7 (GGTGTCAGCGCAAACTGGT) were generated as described previously by co-transfecting low-passage HEK293T cells with lentiviral vector (sgOpti; addgene 85681), packaging vector (psPAX2; addgene 12260), and envelope vector (pVSVg; addgene 138479) using Lipofectamine 2000 (Thermo Fisher) (*Thomas et al., 2022*; *Shalem et al., 2014*). Generated lentiviral particles were used to transduce low-passage BEAS-2B Cas9 and THP-1 Cas9 cells with 200 µg Polybrene (Sigma). Transduced cells were selected for 6 days with 8 µg/ml Puromycin (Sigma). Monoclonal cell lines were generated by limiting dilution and screened by functional IFN-γ ELISpot assay. Genome editing efficiency and clonality were verified by Sanger sequencing and analyzed with the ICE (Synthego) tool (*Conant et al., 2022*). Genomic DNA was isolated from KO clones using the QIAmp DNA Mini Kit (QIAGEN) per the manufacturer's protocol. DNA surrounding sgRNA target region was amplified with Q5 HF DNA polymerase (NEB) and primers (Syt1 F: CCTCAGTAAGTTACACCCTGAC, Syt1 R: TTGTGTCCAGGGTTGCTGTT, Syt7 F: TGAGGGTCTCAAGTTTGGAGG, Syt7 R: TGTCCCCGAATGTGTCCCTA). Sanger sequencing was performed with an Applied Biosystems 3730*xl* 96-capillary DNA Analyzer at the OHSU Vollum Institute DNA Sequencing Core.

## Ribonucleoprotein-mediated gene knockouts

To generate Syt1 and Syt7 KO in the BEAS-2B MR1KO:tetMR1-GFP clone D4 background, a CRISPR Gene Knockout Kit (Synthego) was used per the manufacturer's protocol on CRISPR editing of immortalized cell lines (Synthego). Ribonucleoprotein (RNP) complexes, consisting of three sgRNAs and SpCas9 nuclease, were assembled at a 9:1 ratio in resuspension buffer R. Electroporation of RNPs was performed with the Neon NxT Electroporation System (Invitrogen) with the following conditions: 1400 V, 20 ms, and 2 pulses. After the electroporated cells rested for 3 days, monoclonal cell lines were generated, and genome editing efficiency was analyzed as described above, with the exception of using the primer sets recommended by Synthego.

## Human subjects

This study was conducted according to the principles expressed in the Declaration of Helsinki. All samples were collected with informed consent, and all experiments were conducted according to protocols approved by the Institutional Review Board at Oregon Health & Science University (IRB00000186). Peripheral blood mononuclear cells (PBMCs) were obtained by apheresis from healthy adults and used to expand T cell clones as described below. Human serum was obtained from healthy adults for ELISpot medium as described below. No data is presented on the PBMCs from human subject participants, and sex and gender were not taken into consideration.

## T cell clones

Two T cell clones were previously characterized and used throughout the study: a MAIT cell clone (D426-G11) (*Harriff et al., 2018*) and an HLA-B45-restricted T cell clone (D466-A10) (*Lewinsohn*

*et al., 2007*). For rapid expansion, T cell clones were co-cultured with irradiated allogenic PBMCs and allogenic lymphoblastoid cell lines in RPMI1640 medium (Gibco) containing 10% human serum and anti-CD3 (30 ng/ml, clone OKT3, eBioscience) (*Heinzel et al., 2002*). Recombinant IL-2 (2 ng/ml) was added the following day and every 2–3 days thereafter. T cell clones were washed on Day 5 to remove anti-CD3 and frozen down after at least 11 days. New freeze-back stocks were validated before use by comparing IFN-γ response to a previous freeze back.

## Reagents and chemicals

5-A-RU prodrug (5-A-RU-PABC-Val-Cit-Fmoc, MedChemExpress) was resuspended in DMSO at 10 mM. Doxycycline (Sigma-Aldrich) was resuspended to 2 mg/ml in $H_2O$ and used at 2 µg/ml unless specified. Ac-6-FP (Schirck's laboratories) was resuspended in 0.01 M NaOH at 5.2 mM. Phytohemagglutinin (PHA, Roche) was resuspended at 10 mg/ml in RPMI1640 medium (Gibco) supplemented with 10% human serum, 2% L-glutamine, and 0.1% gentamicin for ELISpot assays. 16% paraformaldehyde (PFA, Electron Microscopy Sciences) was diluted to appropriate concentrations. $CFP10_{2-9}$ peptide was obtained from Genemed Synthesis and resuspended in DMSO at 5 mg/ml.

## ELISpot assays

Multiscreen 96-well mixed cellulose esters plates (MSHAS4510, Millipore) were coated overnight at 4°C with 10 µg/ml anti-IFN-γ antibody (Clone 1-D1K, Mabtech) diluted in coating buffer (0.1 M $Na_2CO_3$, 0.1 M $NaHCO_3$, pH 9.6). Plates were washed three times with sterile PBS (Corning) and incubated with blocking buffer (RPMI (Gibco) containing 10% human serum, 2% L-glutamine, and 0.1% gentamicin (Gibco)) for 1 hr at room temperature. For exogenous antigens and peptide, 1e4 BEAS-2B cells or 1e4 THP-1 cells differentiated with 50 ng/ml phorbol 12-myristate 13-acetate (PMA) (Sigma-Aldrich) for 48 hr prior were plated in duplicate. Cells were incubated with Msmeg supernatant, $CFP10_{2-9}$ peptide, or 5-A-RU prodrug in serial dilutions. PHA (Roche) was used as a positive control. After 1 hr of incubation at 37°C and 5% $CO_2$, 1e4 MAIT cell clone (D426-G11) or 1e4 HLA-B45-restricted T cell clone (D466-A10) were added to the plates based on the antigen. All cells and reagents were resuspended in the blocking buffer described above. Cells were co-cultured for 18 hr at 37°C and 5% $CO_2$. Plates were washed with PBS (Sigma-Aldrich) containing 0.05% Tween-20 (Affymetrix), followed by incubation with ALP-conjugated secondary antibody (Clone 7-B6-1-ALP, Mabtech) diluted in PBS containing 0.5% BSA (Fisher Scientific) and 0.05% Tween-20 for 2 hr at room temperature. Plates were washed again in PBS containing 0.05% Tween-20 and developed using BCIP/NBT-plus developer (Mabtech). IFN-γ spots were enumerated using an AID ELISpot reader and AID EliSpot software (version 7, Autoimmune Diagnostica).

For Mtb infection, 4e5 BEAS-2B cells were infected with H37Rv Mtb (MOI = 8) overnight. 4e5 THP-1 cells differentiated with PMA for 48 hr were washed with PBS and infected with H37Rv Mtb (MOI = 1) for 3 hr. Subsequently, infected cells were serially diluted and plated in duplicate. 1e4 MAIT cell clone (D426-G11) or 1e4 HLA-B45-restricted T cell clone (D466-A10) was added and co-cultured for 18 hr at 37°C and 5% $CO_2$. Plates were processed as described above.

## CFU assays

Cells infected with H37Rv Mtb (MOI = 8) overnight were lysed in ultrapure water the following day. Serial dilutions were plated on 7H10 agar supplemented with glycerol and Middlebrook ADC. Plates were incubated at 37°C and 5% $CO_2$. Triplicates after serial dilutions with PBS + 0.05% Tween-80 were plated and enumerated after 12 days to determine CFU per 100 cells.

## siRNA knockdown

Small-interfering RNAs targeting Syt11 (s23283), Esyt1 (s23607), Esyt2 (s33136), and a missense control (4390844) were obtained from Thermo Fisher Scientific. 1.5e5 BEAS-2B cells were plated in 6-well plates (Corning) and transfected with 50 nM siRNA using Lipofectamine RNAiMAX (Invitrogen) at 80% confluency. Cells were used for ELISpot assays or RNA extraction at 48 hr post-transfection.

## RNA isolation, cDNA synthesis, and qPCR analysis

Total RNA was isolated using the RNeasy Plus Mini Kit (QIAGEN) and reverse transcribed into cDNA using the High-Capacity RNA-to-cDNA Kit (Applied Biosystems) according to the manufacturer's

instructions. Quantitative RT-qPCR was performed on a Step One Plus Real-Time PCR System (Applied Biosystems) using TaqMan Universal PCR Master Mix (Life Technologies). Taqman FAM-MGB probes for *SYT11* (Hs00383056_m1), *ESYT1* (Hs00248693_m1), *ESYT2* (Hs00294020_m1), *SYT1* (Hs00194572_m1), *SYT7 (Hs01590513_m1)*, and *MR1* (Hs00155420_m1) were obtained from Thermo Fisher Scientific. Samples were run in triplicates. Gene expression levels were normalized to *GAPDH* (Hs02758991_g1) of the same sample. Relative expression of the sample was compared to control.

## Flow cytometry assays

For Mtb uptake assays, 4e5 BEAS-2B cells were plated in a 6-well plate and incubated for at least 5 hr at 37°C and 5% $CO_2$. Cells were then infected with mEmeraldRFP-AuxMtb (MOI = 8) overnight. Cells were stained with Live/Dead Fixable Dead Cell Stain Kit (Thermo Fisher) for 20 min on ice, washed, and fixed in 4% PFA. For MR1 surface expression assays, cells were plated in a 12-well plate and treated with 2 µg/ml doxycycline overnight at 37°C and 5% $CO_2$. 10 µM Ac-6-FP or NaOH (solvent control) was added the next day for at least 18 hr. Cells were stained with Live/Dead Fixable Dead Cell Stain Kit for 20 min on ice. Then, cells were stained with APC-conjugated anti-HLA Ia antibody (clone W6/32; BioLegend #311410), APC-conjugated anti-MR1 antibody (clone 26.5; BioLegend #361108), and APC-conjugated isotype control antibody (clone MOPC-173; BioLegend #400222) for 20 min on ice in a FACS buffer (PBS containing 2% human serum, 2% goat serum, and 0.5% FBS). Cells were washed and fixed in 2% PFA. All data were obtained with LSR II (BD) cytometer at the OHSU Flow Cytometry Shared Resource and analyzed with FlowJo software version 10 (TreeStar).

## Flow organellometry assays

WT, Syt1 KO, and Syt7 KO BEAS-2B MR1KO:tetMR1-GFP cells were plated at 9e6 and incubated with 2 µg/ml doxycycline for 5 hr at 37°C and 5% $CO_2$. Cells were infected overnight with AuxMtb (MOI = 10) labeled with Alexa Fluor 647 Succinimidyl Ester (Invitrogen) in the presence of leucine and pantothenate. Postnuclear supernatant was prepared as described previously (*Grotzke et al., 2009*). Post-nuclear supernatant was layered onto 27% Percoll and centrifuged for 1 hr at 4°C in 36,000 × *g* using a 70.1Ti rotor (Beckman). The Percoll gradient was manually fractionated into approximately 50 fractions (200 µl each). Fractions 36–50, the phagosome-containing fractions, were stained with BV421-conjugated anti-CD107a (LAMP-1) antibody (clone H4A3; BioLegend #328625) in FACS buffer and fixed in 2% PFA. All data were obtained with LSR II (BD) cytometer at the OHSU Flow Cytometry Shared Resource and analyzed with FlowJo software version 10 (TreeStar).

## Fluorescence microscopy

All images were acquired on a motorized Nikon TiE stand with a Yokogawa W1 spinning disk unit and a high-powered Agilent laser-emission filter (405–445/50 nm, 488–525/36 nm, and 561–617/73 nm). A 100x (NA 1.49) objective was used and images were captured with an Andor Zyla 5.5 sCMOS camera with 2by2 camera binning. For co-localization between Syt1 and Syt7 with other endosomal markers, 1e5 BEAS-2B cells transfected with Syt1-RFP or Syt7-RFP were plated on 8-well #1.5 glass bottom chamber slides (Nunc) and incubated at 37°C and 5% $CO_2$ overnight. CellLight BacMam 2.0 (Invitrogen) for early endosomes (Rab5a-GFP, C10586), late endosomes (Rab7-GFP, C10588), or lysosomes (LAMP1-GFP, C10596) was added the next day and incubated overnight. For co-localization between Syt1 and Syt7 with MR1, 2–2.5e5 polyclonal BEAS-2B:tetMR1-GFP cells transfected with Syt1-RFP or Syt7-RFP were plated and incubated with 2 µg/ml doxycycline at 37°C and 5% $CO_2$ overnight. To measure changes in MR1 cellular distribution in Syt1 and Syt7 KO BEAS-2B MR1KO:tetMR1-GFP clone D4 cells, 2e5 cells were plated and incubated with 2 µg/ml doxycycline at 37°C and 5% $CO_2$. After 4 hr, either CellLight BacMam 2.0 (Invitrogen) for Rab5a-RFP (C10587), Rab7-RFP (C10589), and LAMP1-RFP (C10597) or 10 µM Ac-6-FP and NaOH (solvent control) were added to quantify co-localization with endosomal compartments or to induce MR1 translocation to the cell surface. For AuxMtb infection, 2–2.5e5 cells were plated on 4-well #1.5 glass bottom chamber slides (Nunc) and incubated with 2 µg/ml doxycycline for 4 hr at 37°C and 5% $CO_2$. Cells were infected overnight with mEmeraldRFP-AuxMtb or AuxMtb (MOI = 5) labeled with Alexa Fluor 555 Succinimidyl Ester (Invitrogen) in the presence of leucine and pantothenate. Before imaging, to ensure randomization, cells were stained with or without NucBlue Live ReadyProbes (Invitrogen) and imaged in an unbiased manner based on their NucBlue, RFP, or GFP expression as appropriate.

## Image analysis

Co-localizations were analyzed using the 'Spots' function and 'spots colocalization' MatLab Xtension module on Imaris 7 (Bitplane). Area of MR1 vesicles was classified into small (≤1 $\mu m^2$) and large vesicles (>1 $\mu m^2$) using the 'Surfaces' function based on MR1-GFP fluorescence, using segmentation setup with Surfaces Detail 0.260 $\mu m$, Absolute Intensity Thresholding method, and Classification using surface area. For each cell, areas of small and large MR1 vesicles were averaged in Excel. Surface overlaps between MR1 and AuxMtb for each cell were analyzed using the 'Surfaces' function similar to above and calculated 'overlapped area ratio to surfaces' under detailed specific values using Imaris. Total number and average speed of MR1 vesicles for each cell were analyzed using the 'Spots' function. Number of MR1 vesicles within 1 $\mu m$ of Mtb was enumerated by first defining MR1-GFP vesicles as 'Spots' and AuxMtb-RFP as 'Surfaces'. Then, MR1 vesicles were classified into two groups (≤1 and >1 $\mu m$) using 'shortest distance to surface' with a classification filter. All analysis involving spots, surfaces, and classifications was done using Imaris 7 and 10 (Bitplane).

## Statistical analysis

All data were analyzed with Prism 10 (GraphPad). At least three independent experiments were performed and plotted as mean ± SEM. Statistical significance for ELISpot assays was determined by non-linear regression using [Agonist] vs. response model with three parameters. Best-fit values of top and $EC_{50}$ parameters were compared between the curves, and p-values were calculated using extra sum-of-squares $F$ test. No constraints were applied to the bottom and top parameters, and the $EC_{50}$ parameter was constrained to be greater than 0 for p-value calculation. Statistical significances for microscopy and others were determined by two-tailed unpaired $t$-test (for two groups), a one-way ANOVA with Dunnett's multiple comparisons test (for three groups), or a two-way ANOVA with Sidak's or Dunnett's multiple comparisons tests was conducted for statistical analysis. A p-value of <0.05 was considered statistically significant.

## Materials availability

All cell lines and plasmids are available upon request and completion of a Material Transfer Agreement.

# Acknowledgements

We acknowledge expert technical assistance by staff in the Advanced Light Microscopy Core (RRID:SCR_009961) in the Department of Neurology and Jungers Center at Oregon Health and Science University. We thank staff at the Vollum DNA Sequencing Core. Analytical flow cytometry was performed in the OHSU Flow Cytometry Shared Resource (RRID:SCR_009974). We also acknowledge the assistance of the Oregon Clinical & Translational Research Institute, which is supported by the National Center for Advancing Translational Sciences, National Institutes of Health, through Grant Award Number UL1TR002369. The contents do not represent the views of the U.S. Department of Veterans Affairs or the United States Government. Lastly, we are grateful to Dr. William Jacobs for sharing the Mtb auxotroph and Dr. Shogo Soma for generation of mEmeraldRFP-AuxMtb. This work was supported by NIH T32HL083808 (SK), R21AI151079 (EK), K08AI153359 (EK), and U.S. Department of Veterans Affairs Merit Award I01BX000533 (DML).

# Additional information

## Funding

| Funder | Grant reference number | Author |
|---|---|---|
| National Heart Lung and Blood Institute | T32HL083808 | Se-Jin Kim |
| National Institute of Allergy and Infectious Diseases | R21AI151079 | Elham Karamooz |
| National Institute of Allergy and Infectious Diseases | K08AI153359 | Elham Karamooz |

| Funder | Grant reference number | Author |
| --- | --- | --- |
| United States Department of Veterans Affairs | I01BX000533 | David Lewinsohn |
| National Center for Advancing Translational Sciences, National Institutes of Health | UL1TR002369 | Se-Jin Kim<br>Jessie C Peterson<br>Fikadu G Tafesse<br>Corinna A Kulicke<br>Elham Karamooz<br>David Lewinsohn |

The funders had no role in study design, data collection, and interpretation, or the decision to submit the work for publication.

## Author contributions

Se-Jin Kim, Conceptualization, Data curation, Formal analysis, Funding acquisition, Validation, Investigation, Visualization, Methodology, Writing – original draft, Project administration, Writing – review and editing; Jessie C Peterson, Data curation, Formal analysis, Investigation, Methodology, Writing – original draft, Writing – review and editing; Andrew J Olive, Fikadu G Tafesse, Corinna A Kulicke, Formal analysis, Investigation, Methodology, Writing – original draft, Writing – review and editing; Elham Karamooz, Conceptualization, Resources, Data curation, Formal analysis, Supervision, Funding acquisition, Validation, Investigation, Methodology, Writing – original draft, Project administration, Writing – review and editing; David Lewinsohn, Conceptualization, Resources, Data curation, Formal analysis, Supervision, Funding acquisition, Investigation, Methodology, Writing – original draft, Project administration, Writing – review and editing

## Author ORCIDs

Se-Jin Kim ⓘ https://orcid.org/0000-0002-6704-4644
Andrew J Olive ⓘ https://orcid.org/0000-0003-3441-3113
Fikadu G Tafesse ⓘ https://orcid.org/0000-0002-8575-4164
Corinna A Kulicke ⓘ https://orcid.org/0000-0002-4217-3095
Elham Karamooz ⓘ https://orcid.org/0000-0002-7185-5490
David Lewinsohn ⓘ https://orcid.org/0000-0001-9906-9494

## Ethics

This study was conducted according to the principles expressed in the Declaration of Helsinki. All samples were collected with informed consent, and all experiments were conducted according to protocols approved by the Institutional Review Board at Oregon Health & Science University (IRB00000186).

Reviewer #1 (Public review): https://doi.org/10.7554/eLife.108318.3.sa1
Reviewer #3 (Public review): https://doi.org/10.7554/eLife.108318.3.sa2
Author response https://doi.org/10.7554/eLife.108318.3.sa3

# Additional files

## Supplementary files

Supplementary file 1. Microscopy image analysis.
Supplementary file 2. Plasmid sequences of Syt1-RFP and Syt7-RFP.
MDAR checklist

## Data availability

All relevant data can be found within the article and its supplementary information.

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
