## [Editor Report · eLife Assessment]

This **useful** study examines the contribution of synaptotagmin 1 and synaptotagmin 7 to metabolite antigen presentation to mucosal-associated invariant T (MAIT) cells; it begins to address a critical gap in our understanding of the antigen presentation mechanisms of these cells. Strengths of the study include the use of Mtb to study the dynamics of antigen presentation to MAIT cells instead of a synthetic antigen. The strength of the evidence to support the conclusion is **solid**.

---

## [Referee Report · Reviewer #1 (Public review)]

Summary:

Synaptotagmin (Syt) 1 and Syt7 specifically promote (are critical for) MAIT cell activation in response to M.tb-infected bronchial epithelial cell line BEAS-2B (Fig. 1) and monocyte-like cell line THP-1 (Fig. 3), but not at the M.smeg-infected conditions. Esyt2 shows a similar effect. This work also displayed co-localization of Syt1 and Syt7 with Rab7a and Lamp1, but not with Rab5a (Fig. 5). Loss of Syt1 and Syt7 resulted in a larger area of MR1 vesicles (Fig. 6f) and an increased number of MR1 vesicles in close proximity to an Auxotrophic Mtb-containing vacuoles during infection (Fig. 7ab). Moreover, flow organellometry to separate phagosomes from other subcellular fractions and identify enrichment of auxotrophic Mtb-containing vacuoles in fractions 42-50, which were enriched with Lamp1+ vacuoles or phagosomes (Fig.7e-f).

Strengths:

This work convincingly associated Syt1 and Syt7 with late endocytic compartments and Mtb+ vacuoles. Gene editing of Syt1 and Syt7 loci of bronchial epithelial and monocyte-like cells supported Syt1 and Syt7 facilitated maintaining a normal level of antigen presentation for MAIT cell activation in Mtb infection. Imaging analyses provided solid evidence to support that Syt1 and Syt7 mutants enhanced the size of MR1-resided vesicles, the overlaps of MR1 with M.tb fluorescent signal, and the MR1 proximity with Mtb-infected vacuoles, suggesting that Syt1 and Syt7 proteins help antigen presentation for MAIT activation in Mtb infection.

Weaknesses:

Current data could be improved to support the conclusion that "This study identifies a pathway in which Syt1 and Syt7 facilitate the translocation of MR1 from Mtb-containing vacuoles, potentially to the cell surface for antigen presentation". Likewise, the current data are more supportive of a different conclusion.

Comments on revisions:

Authors have been very responsive to the review comments, except for keeping a very strong conclusion. Suggest rewriting the conclusions "identifies a specialized pathway", "facilitate the translocation", "from Mtb-containing vacuoles", and "potentially to the cell surface" to be more reflective of the data.

---

## [Referee Report · Reviewer #3 (Public review)]

Summary:

In the submitted manuscript the authors investigate the role of Synaptotagmins (Syt1) and (Syt7) in MR1 presentation of Mtb antigens. By using Syt1 and Syt7 knock down the authors determine that these molecules are required to effectively control Mtb infection.

Strengths:

In the first series of experiments, the authors determined that knocking down Syt1 and Sy7 in antigen-presenting cells decreases IFN-γ production following cellular infection with Mtb. These experiments are well performed and controlled.

Comments on revisions:

The revised manuscript offers further support to the role of Synaptogamins 1 and 7 in MR1 trafficking during MT infection

---

## [Author Response]

The following is the authors’ response to the original reviews.

**Public Reviews:**

**Reviewer #1 (Public review):**
Summary:The manuscript "Synaptotagmin 1 and Synaptotagmin 7 promote MR1-mediated presentation of *Mycobacterium tuberculosis* antigens", authored by Kim et al., showed that the calcium-sensing trafficking proteins Synaptotagmin (Syt) 1 and Syt7 specifically promote (are critical for) MAIT cell activation in response to Mtb-infected bronchial epithelial cell line BEAS-2B (Fig. 1) and monocyte-like cell line THP-1 (Figure 3) . This work also showed co-localization of Syt1 and Syt7 with Rab7a and Lamp1, but not with Rab5a (Figure 5). Loss of Syt1 and Syt7 resulted in a larger area of MR1 vesicles (Figure 6f) and an increased number of MR1 vesicles in close proximity to an Auxotrophic Mtb-containing vacuoles during infection (Figure 7ab). Moreover, flow organellometry was used to separate phagosomes from other subcellular fractions and identify enrichment of auxotrophic Mtb-containing vacuoles in fractions 42-50, which were enriched with Lamp1+ vacuoles or phagosomes (Figures 7e-f).Strengths:This work nicely associated Syt1 and Syt7 with late endocytic compartments and Mtb+ vacuoles. Gene editing of Syt1 and Syt7 loci of bronchial epithelial and monocyte-like cells supported Syt1 and Syt7 facilitated maintaining a normal level of antigen presentation for MAIT cell activation in Mtb infection. Imaging analyses further supported that Syt1 and Syt7 mutants enhanced the overlaps of MR1 with Mtb fluorescence, and the MR1 proximity with Mtb-infected vacuoles, suggesting that Syt1 and Syt7 proteins help antigen presentation in Mtb infection for MAIT activation.Weaknesses:Additional data are needed to support the conclusion, "identify a novel pathway in which Syt1 and Syt7 facilitate the translocation of MR1 from Mtb-containing vacuoles" and some pieces of other evidence may be seen by some to contradict this conclusion.

We thank the reviewer for their positive and constructive comments. Because MR1 presents small molecule metabolites, specifically identifying MR1 molecules loaded with antigens derived from intracellular Mtb infection remains a significant technical challenge. Therefore, we agree that some of our approaches measure antigen-loaded MR1 indirectly. For example, IFN-γ release from a MAIT cell clone serves as a sensitive surrogate readout for the presence of antigen-loaded MR1 at the cell surface. This has been demonstrated in previous work showing that IFN-γ release from MAIT cells correlated with loaded MR1 molecules measured using flow cytometry and a TCR based tetramer (Kulicke et al., 2024). In this context, Syt1 and Syt7 represent the first endosomal trafficking proteins we have identified that play a specific role in MR1-mediated presentation of Mtb-derived metabolites. Syt1 and Syt7 do not contribute to the presentation of an exogenously delivered MR1 ligands, such as Ac-6-FP loaded in the ER or *M. smegmatis* supernatant. In Syt1 and Syt7 knockout cells expressing MR1-GFP, larger MR1 vesicles are observed, but MR1 continues to co-localize with LAMP1 similar to wildtype cells. Furthermore, Syt1 and Syt7 knockout cells exhibit an increased number of MR1 vesicles near the Mtb-containing vacuoles compared to wildtype cells. To increase the statistical power of our microscopy analyses, we have analyzed additional cells. Although the absolute magnitude of the observed effects is modest, T cell activation is highly sensitive to the number of loaded antigen presenting molecules at the cell surface. Also, a complementary approach using flow organellometry confirmed increased MR1 expression within Mtb^+^LAMP1^+^ vesicles in Syt7 knockout cells. Thus, these findings suggest a mechanism whereby Syt1 and Syt7 facilitate the trafficking of loaded MR1 molecules from the Mtb-containing vacuoles to the plasma membrane. This specialized mechanism may be analogous to the previously described role of Syt7 in MHC class II trafficking (Becker et al., 2009). In our model, we observed increased accumulation and expression of MR1 within Mtb-containing vacuoles in Syt7 knockout cells.

**Reviewer #2 (Public review):**
Summary:The study demonstrates that calcium-sensing trafficking proteins Synaptotagmin (Syt) 1 and Syt7 are involved in the efficient presentation of mycobacterial antigens by MR1 during *M. tuberculosis* infection. This is achieved by creating antigen-presenting cells in which the Syt1 and Syt7 genes are knocked out. These mutated cell lines show significantly reduced stimulation of MAIT cells, while their stimulation of HLA class I-restricted T cells remains unchanged. Syt1 and Syt7 co-localize in a late endo-lysosomal compartment where MR1 molecules are also located, near *M. tuberculosis*-containing vacuoles.Strengths:This work uncovers a new aspect of how mycobacterial antigens generated during infection are presented. The finding that Syt1 and Syt7 are relevant for final MR1 surface expression and presentation to MR1-restricted T cells is novel and adds valuable information to this process. The experiments include all necessary controls and convincingly validate the role of Syt1 and Syt7. Another key point is that these proteins are essential during infection, but they are not significant when an exogenous synthetic antigen is used in the experiments. This emphasizes the importance of studying infection as a physiological context for antigen presentation to MAIT cells. An additional relevant aspect is that the study reveals the existence of different MR1 antigen presentation pathways, which differ from the endoplasmic reticulum or endosomal pathways that are typical for MHC-presented peptides.Weaknesses:The reduced MAIT cell response observed with Syt1 and Syt7-deficient cell lines is statistically significant but not completely abolished. This may suggest that only some MR1-loaded molecules depend on these two Syt proteins. Further research is needed to determine whether, during persistent *M. tuberculosis* infection, enough MR1-loaded molecules are produced and transported to the plasma membrane to sufficiently stimulate MAIT cells. The study proposes that other Syt proteins might also play a role, as outlined by the authors. However, exploring potential redundant mechanisms that facilitate MR1 loading with antigens remains a challenging task.

We appreciate the reviewer’s comments and feedback. Syt1 and Syt7 knockout cells do not completely abolish MR1-mediated presentation of Mtb-derived metabolites. We agree that the likely explanation is that there are redundancies within the antigen presentation pathways. Whether these redundancies are due to other endosomal trafficking proteins or other intracellular compartments where MR1 loading can occur remains unknown. Moreover, Mtb-derived antigens can access the ER, where Syt1 and Syt7 are not involved, thereby enabling an ER-mediated pathway for MR1 antigen presentation. It is also important to note that relatively few (<10) loaded MHC class I molecules are sufficient to trigger T cell activation (Brower et al., 1994; Sykulev et al., 1995; Sykulev et al., 1996). A major challenge in exploring these mechanisms is due to the inability to directly track small molecule Mtb-derived antigens as they are loaded onto MR1 and presented at the cell surface. These hurdles are briefly outlined in the discussion as future directions. Nonetheless, Syt1 and Syt7 are the first endosomal trafficking proteins identified to have a specific effect on MR1-mediated presentation of Mtb-derived antigens.

**Reviewer #3 (Public review):**
Summary:In the submitted manuscript, the authors investigate the role of Synaptotagmins (Syt1) and (Syt7) in MR1 presentation of MtB.Strengths:In the first series of experiments, the authors determined that knocking down Syt1 and Sy7 in antigenpresenting cells decreases IFN-γ production following cellular infection with Mtb. These experiments are well performed and controlled.Weaknesses:Next, they aim to mechanistically investigate how Syt1 and Syt7 affect MtB presentation. In particular, they focus on MR1, a non-classical MHC-I molecule known to present endogenous and exogenous metabolites, including MtB metabolites. Results from these next series of experiments are less clear. Firstly, they show that knocking down Syt1 and Sy7 does not change MtB phagocytosis as well as MR1 ER-plasma membrane translocation. Based on this, they suggest that Syt1 and Syt7 may affect MR1 trafficking in endosomal compartments. However, neither subcellular compartment analysis nor flow organelleometry clearly establishes the role of Syt1 and Syt7 in MtB trafficking. Altogether, the notion that Synaptotagmins facilitate MR1 interaction with Mtb-containing compartments and its vesicular transport was already known. As such, the manuscript should add additional insight on where/how the interaction occurs. The reviewer is left with the notion that Syt1 and Sy7 may affect MR1 presentation, facilitating the trafficking of MR1 vesicles from endosomal compartments to either the cell surface or other endosomal compartments. The analysis is observational and additional data or discussion could address what the insight gained beyond what is already known from the literature.

We thank Reviewer 3 for their comments. Our hypothesis is that Syt1 and Syt7 mediate MR1 trafficking rather than Mtb trafficking. While Syt7 has previously been implicated in MHC class II trafficking and vesicular transport, this study is the first to explore in detail the roles of Syt1 and Syt7 in MR1-mediated presentation of Mtb-derived metabolites. Since current technologies do not allow direct tracking of Mtbderived antigens loaded onto MR1, we relied on complementary approaches including IFN-γ release from MAIT cells, flow cytometry, fluorescence microscopy, and flow organelleometry. Both flow organelleometry and fluorescence microscopy show increased MR1 expression at Mtb-containing vacuoles in Syt7 knockout cells. Since total MR1 expression measured by flow cytometry and the overall number of MR1 vesicles remain unchanged, these data support a mechanism in which Syt7 facilitates the trafficking of antigen-loaded MR1 from Mtb-containing vacuoles to the cell surface, consistent with the observed reduction in MAIT cell IFN-γ release.

**Recommendations for the authors:**

**Reviewer #1 (Recommendations for the authors):**
Concern 1, the data in the current manuscript have not been sufficient to "identify a novel pathway in which Syt1 and Syt7 facilitate the translocation of MR1 from Mtb-containing vacuoles, potentially to the cell surface for antigen presentation" (Last part of Abstract). To conclude this, additional pieces of data are needed: (a) Mtb-containing vacuoles associate with MR1 protein expression; (b) MR1+ vesicles traffic from one subcellular location to another; (c) Syt1 or Syt7 KO reduces MR1 vesicles at a downstream subcellular location, e.g., the cell surface. Important evidence supporting the "facilitation of translocation" is missing on whether Syt1 or Syt7 KO reduces MR1 vesicle traffic from one location to another.

We thank the reviewer for their detailed suggestions to improve our proposed model. We would like to clarify that Figure 7g demonstrates increased MR1 protein expression in Syt7 knockout cells, as assessed by flow organellometry. This approach allowed us to specifically distinguish AuxMtb^+^LAMP1^+^ compartments (Mtb-containing vacuoles) and to quantify MR1 expression using geometric mean fluorescence intensity. Moreover, in both Syt1 and Syt7 knockout cells, MR1+ vesicles are retained within lysosomal compartments, characterized by vesicle enlargement and accumulation. Therefore, we did not observe trafficking of MR1+ vesicles to other subcellular locations or to the plasma membrane. A key limitation, however, is the lack of current technologies that allow direct measurement of MR1 surface expression specifically during intracellular Mtb infection via flow cytometry. Given this limitation, IFN-γ ELISpot is a sensitive surrogate and supports the conclusion that loss of Syt1 and Syt7 results in decreased MR1 presentation of Mtb-derived antigens at the plasma membrane.

The results "a significant increase in the number of MR1 vesicles within 1 μm of AuxMtb for Syt1 (1.13 {plus minus} 0.46) and Syt7 KO (1.31 {plus minus} 0.46) cells compared to WT cells (Fig.7b)." and "the surface of MR1 vesicles in Syt1 and Syt7 KO cells showed a 3-fold increase in overlap area with Mtb surfaces (Fig.7d)." may need to be further elaborated on whether MR1+vacuoles and Mtb+ vacuoles are overlapped or are adjacent. Figure 7b shows several groups of vacuoles with the same distance. This needs a larger sample size to randomize this distance measurement, for example, calculating 50~100 Mtb+ vacuoles.

We appreciate the reviewer’s critical comments and suggestions. To quantify distance and surface overlap, the microscopy images were acquired from a single optical plane rather than full z-stacks. As a result, it is not possible to definitively determine whether MR1+ vesicles and Mtb-containing vacuoles are directly overlapping or adjacent. In response to the reviewer’s suggestion, we increased the sample size for both distance (n=51-53) and surface overlap analyses (n=51-53). Using the larger sample size, we observed a significant increase in the number of MR1 vesicles located within 1μm of AuxMtb in both Syt1 (1.23±0.21) and Syt7 knockout (1.28±0.22) cells. Also, there was an approximately 4-fold increase in MR1-Mtb surface overlap area compared to wildtype cells.

Results from "performed flow organellometry to separate phagosomes from other subcellular fractions and identified enrichment of Mtb-containing vacuoles in fractions 42-50 (Fig.7e-f)" could not distinguish the difference between WT and Syt1/Syt7 KO, or further support the role of Syt1/Syt7 in endocytic trafficking. More specifically, authors claimed that "enhanced MR1 expression in Mtb+LAMP1+ compartments via flow organellometry in Syt1 and Syt7 KO cells.", may not be supported by Figure 7f, which does not show a difference in MR1 expression between Syt1 KO or Syt7 KO and WT.

We appreciate the reviewer’s concerns and would like to clarify the interpretation of Figures 7f and 7g. Figure 7f demonstrates: (a) enrichment Mtb-containing vacuoles within fractions 42-50, (b) coenrichment of LAMP1+ vesicles within these Mtb-containing fractions, and (c) comparable subcellular fractionation profiles across wildtype, Syt1 knockout, and Syt7 knockout cells, indicating no major differences in fraction distribution. Differences in MR1 expression are shown in Figure 7g, which compares MR1 expression as the geometric mean fluorescence intensity within the fraction exhibiting the highest percentage of AuxMtb^+^LAMP1^+^ across all fractions. We observed significant increase in MR1 expression in Syt7 knockout cells compared to wildtype cells.

Concern 2, in abstract, "Loss of Syt1 and Syt7 results in enlarged MR1 vesicles and an increased number of MR1 vesicles in close proximity to Mtb-containing vacuoles during infection.". Although numbers of MR1 vesicles within 1um of Mtb increase (Figure 7b) and areas of MR1+ vacuoles for WT and KO cells enhance (Figure 6f), but numbers of MR1 vesicles/cells are not different between WT and Syt1 and Sy7 KO (Fig. 7c). These imaging analyses, including other figure panels, need more explicit presentation of (most if not all) random images for calculation, annotation of MR1-vacuoles for calculation, and raw statistical data for mean and p value calculation. These raw data can be presented in supplemental figure panels.

We thank the reviewer for these suggestions. We have included more details on randomization, technical procedures, and statistical analyses in methods section for “Fluorescence Microscopy,” “Image Analysis,” and “Statistical Analysis.” Raw data collection and statistical data are presented in the supplemental data.

Concern 3, additional evidence that does not support the conclusion "This study identifies a novel pathway in which Syt1 and Syt7 facilitate the translocation of MR1 from Mtb-containing vacuoles" (the last part of Abstract). This additional unsupportive evidence includes: (a) MR1 expression on the cell surface is not impacted or not different among WT, Syt1 KO, and Syt7 KO of BEAS-2B cells (Fig. 6d). (b) "Live-cell imaging showed no differences in MR1 cellular distribution in the presence or absence of Ac-6FP between WT, Syt1, and Syt7 KO BEAS-2B:TET-MR1GFP cells as MR1 translocated from the ER and vesicles to the cell surface as expected (Figure 6c).

We thank the reviewer for this comment and would like to clarify our use of Ac-6-FP. Figures 6c and 6d examine MR1 cellular distribution and surface expression in the presence or absence of Ac-6-FP. Ac-6-FP is a small MR1 ligand that is loaded in the ER and promotes MR1 surface stabilization and trafficking to the cell membrane. In contrast, Mtb primarily resides within membrane-bound phagosomes. MR1 presentations of soluble/exogenously delivered ligands versus intracellular Mtb-derived antigens have shown to involve distinct pathways and endosomal trafficking proteins (Harriff et al., 2016; Karamooz et al., 2019; Karamooz et al., 2025). Findings from Figures 6c and 6d show that Syt1 and Syt7 do not contribute to the presentation of small soluble and ER-loaded ligands such as Ac-6-FP. Instead, they specifically contribute in MR1 presentation of Mtb-derived metabolites by translocating MR1 from Mtbcontaining vacuoles in the context of intracellular Mtb infection

Other concerns:(1) Figure 1a uses Ct value to measure Syt1 and Syt7 expression levels, but a comparison with GAPDH Ct cycle numbers in different cell types will be helpful for understanding.

We appreciate the reviewer’s suggestion of including GADPH Ct cycle numbers. We have revised Figure 1a to show Ct values for Syt1, Syt7, and GAPDH in both BEAS-2B and THP-1 cells.

(2) Figure 1b indel, shown with an ICE method, should be confirmed with protein expression levels to interpret functional results.

We thank the reviewer for raising this concern. We attempted to assess protein levels by western blot using multiple antibodies from both Abcam and Synaptic Systems. However, we were unable to identify a suitable antibody that reliably detected endogenous Syt1 or Syt7 protein levels.

(3) Figure 1c. HLA-B45-restricted T cell clones also show some marginal reduction of IFN-γ spot responses and are more different in Figure 6b. Please discuss this conflicting data. Also, need a reference to support whether the exogenous CFP peptide antigen is presented via surface or endocytic antigen loading.

We agree with the reviewer that there are some marginal reductions of IFN-γ responses for HLA-B45restricted T cell clones. Since T cell clones are used from frozen, there can be differences in maximal responses between T cell clones and expansions of the same T cell clone. However, the comparisons include a control arm and pool data from multiple experiments to reach statistical power and validity. In addition, Figure 6b shows Syt1 and Syt7 KO cells in the background of BEAS-2B MR1KO:tetMR1-GFP clone D4 cells, which overexpresses MR1 that may contribute to variability and potentially account for the observed differences. With respect to exogenous CFP peptide loading, earlier studies on peptides and antigen presenting cells demonstrated that peptides can be loaded onto fixed cells and subsequently presented to T cells (Shimonkevitz et al., 1983; Watts et al., 1985). Based on these findings, it is reasonable to assume that substantial peptide exchange occurs at the cell surface when exogenous peptides are added to antigen presenting cells.

(4) Figure 2e: Delta CT values of Syt1, Syt7 in WT, KO cells can be shown together with Ct values of GAPDH or B2m house-keeping genes to help readers determine the efficiency of Syt1 and 7 mutation at the gene expression level. Also, in Figure 4a, the baseline of Ct values for GAPDH can be plotted together.

As suggested by the reviewer, we have revised Figure 2e and 4a to include CT values for the genes of interest as well as housekeeping gene GAPDH.

(5) Figure 3c and Figure 1d: M.smeg infection can be shown to be more comparable with Mtb infection.

We thank the reviewer for this thoughtful comment. Although *M. smegmatis* infection could serve as a comparable control, *M. smegmatis* secretes large amounts of MR1 ligands derived from riboflavin metabolism. This makes it difficult to distinguish between extracellular and intracellular antigens, and to directly compare with Mtb infection, which is specifically an intracellular infection model.

(6) Figure 4e: It appears Esyt2 Knockdown shows strong inhibition of MAIT activation mediated by BEAS2B cells with Mtb infection and M.smeg supernatant stimulation. Please add other relevant data, such as MR1 cell surface expression and colocalization, and discuss these results with Syt proteins.

We appreciate the reviewer’s suggestion to include relevant data for Esyt2 knockdown. We performed flow cytometry analysis of Esyt2 knockdown cells and found surface MR1 expression under basal conditions. Treatment with Ac-6-FP resulted in increased MR1 surface stabilization, but MR1 surface level was significantly lower than those observed in missense control cells. Therefore, Esyt2 is not specific to MR1 presentation of Mtb-derived metabolites and instead may play a broader role in overall MR1 antigen presentation, including intracellular Mtb-derived antigens, exogenous antigens, and ER-loaded Ac-6-FP.

(7) Figure 5 colocalization computational analyses can be more explicitly presented regarding randomization, technical procedures, and statistical analyses, as stated in Concern 2.

As suggested, we have included more details in methods section and added the supplemental data.

(8) Figure 6a: Syt1 and Syt7 protein expressions are also suggested to confirm the mutation, similar to the confirmation for Figures 1 and 3.

We thank the reviewer for raising this concern. As discussed previously, we have not identified a suitable antibody for human Syt1 and Syt7. We have tested multiple antibodies from Abcam and Synaptic Systems.

(9) For statistical analyses, "non-linear regression analysis comparing best-fit values of top and EC50 were used to calculate p-values by extra sum-of-squares F test" (Figure 6b) and "non-linear regression analysis of pairwise comparison to WT on best-fit values of top and EC50 were used to calculate p-values by extra sum-of-squares F test." (Figure 3bc), readers may need more specific demonstration in supplemental figures on how statistical analyses have been performed.

We appreciate the reviewer’s suggestion to include more detailed information regarding the statistical analyses. For clarification, data presented in Figures 6b and 3bc were analyzed using the same statistical analysis in Prism 10. Specifically, nonlinear regression (curve fit) was performed using the [Agonist] vs. response model with three parameters. Best-fit values for the top and EC_50_ parameters were compared using an extra sum-of-squares F test.No constraints were applied to the bottom and top parameters, and the EC_50_ parameter was constrained to be greater than 0 for p-value calculation. We have revised the Statistical Analysis section of the Methods to more clearly describe this approach.

(10) In discussion, the background section for Syt1 and Syt7 is more appropriate to be in the introduction. This will allow readers to better understand the association of Syt proteins with MR1 and the necessity to study the impact of Syt on MR1 trafficking.

We thank the reviewer for this suggestion. We believe that the basic background and relevance of Syt1 and Syt7 in MR1 trafficking are covered in the introduction; however, we have added details to help readers understand their impact.

**Reviewer #2 (Recommendations for the authors):**
This reviewer has no requests for implementation and congratulates the authors on this nice piece of work.

We thank the reviewer for the positive comments.

**Reviewer #3 (Recommendations for the authors):**
Complete trafficking experiments to pinpoint the trafficking relationship between Syt 1 and 7 and MR1 in MtB infection.

We appreciate the reviewer’s insightful comment. As this study represents the first detailed investigation into the roles of Syt1 and Syt7 in MR1-mediated presentation of Mtb-derived metabolites, we agree that a fully resolved trafficking mechanism has not yet been established. A major limitation is the inability to directly track Mtb-derived antigens as they are loaded onto MR1 and trafficked to the cell surface. Therefore, we relied on complementary functional and microscopy-based approaches, including IFN-γ ELISpot assays, flow cytometry, fluorescence microscopy, and flow organellometry, to infer the trafficking relationships between Syt1, Syt7, and MR1 during intracellular Mtb infection. Our data support a model that Syt1 and Syt7 facilitates the trafficking of MR1 from Mtb-containing vacuoles to the plasma membrane. This interpretation is supported with the increased accumulation of MR1 in Mtb-containing vacuoles and reduction in MAIT cell IFN-γ release observed in Syt1 and Syt7 knockout cells.

References

(1) Becker, S. M., Delamarre, L., Mellman, I., & Andrews, N. W. (2009). Differential role of the Ca(2+) sensor synaptotagmin VII in macrophages and dendritic cells. Immunobiology, 214(7), 495–505.

(2) Brower, R. C., England, R., Takeshita, T., Kozlowski, S., Margulies, D. H., Berzofsky, J. A., & Delisi, C. (1994). Minimal requirements for peptide-mediated activation of CD8+ CTL. Molecular immunology, 31(16), 1285–1293.

(3) Harriff, M. J., Karamooz, E., Burr, A., Grant, W. F., Canfield, E. T., Sorensen, M. L., Moita, L. F., & Lewinsohn, D. M. (2016). Endosomal MR1 Trafficking Plays a Key Role in Presentation of *Mycobacterium tuberculosis* Ligands to MAIT Cells. PLoS pathogens, 12(3), e1005524.

(4) Karamooz, E., Harriff, M. J., Narayanan, G. A., Worley, A., & Lewinsohn, D. M. (2019). MR1 recycling and blockade of endosomal trafficking reveal distinguishable antigen presentation pathways between *Mycobacterium tuberculosis* infection and exogenously delivered antigens. Scientific reports, 9(1), 4797.

(5) Karamooz, E., Kim, S. J., Peterson, J. C., Tammen, A. E., Soma, S., Soll, A. C. R., Meermeier, E. W., Khuzwayo, S., & Lewinsohn, D. M. (2025). Two-pore channels in MR1-dependent presentation of *Mycobacterium tuberculosis* infection. PLoS pathogens, 21(8), e1013342.

(6) Kulicke, C. A., Swarbrick, G. M., Ladd, N. A., Cansler, M., Null, M., Worley, A., Lemon, C., Ahmed, T., Bennett, J., Lust, T. N., Heisler, C. M., Huber, M. E., Krawic, J. R., Ankley, L. M., McBride, S. K., Tafesse, F. G., Olive, A. J., Hildebrand, W. H., Lewinsohn, D. A., Adams, E. J., … Harriff, M. J. (2024). Delivery of loaded MR1 monomer results in efficient ligand exchange to host MR1 and subsequent MR1T cell activation. Communications biology, 7(1), 228.

(7) Shimonkevitz, R., Kappler, J., Marrack, P., & Grey, H. (1983). Antigen recognition by H-2restricted T cells. I. Cell-free antigen processing. The Journal of Experimental Medicine, 158(2), 303–316.

(8) Sykulev, Y., Cohen, R. J., & Eisen, H. N. (1995). The law of mass action governs antigen-stimulated cytolytic activity of CD8+ cytotoxic T lymphocytes. Proceedings of the National Academy of Sciences of the United States of America, 92(26), 11990–11992.

(9) Sykulev, Y., Joo, M., Vturina, I., Tsomides, T. J., & Eisen, H. N. (1996). Evidence that a single peptide-MHC complex on a target cell can elicit a cytolytic T cell response. Immunity, 4(6), 565– 571.

(10) Watts, T. H., Gariépy, J., Schoolnik, G. K., & McConnell, H. M. (1985). T-cell activation by peptide antigen: effect of peptide sequence and method of antigen presentation. Proceedings of the National Academy of Sciences of the United States of America, 82(16), 5480–5484.